# Toll/interleukin-1 receptor (TIR) domain-containing proteins have NAD-RNA decapping activity

Xufeng Wang [1,2,3,9], Dongli Yu [4,5,6,9], Jiancheng Yu[7], Hao Hu[1,2,3], Runlai Hang[1,2,3], Zachary Amador [3], Qi Chen [7,8], Jijie Chai[4,5] & Xuemei Chen [1,2] ✉

The occurrence of NAD$^+$ as a non-canonical RNA cap has been demonstrated in diverse organisms. TIR domain-containing proteins present in all kingdoms of life act in defense responses and can have NADase activity that hydrolyzes NAD$^+$. Here, we show that TIR domain-containing proteins from several bacterial and one archaeal species can remove the NAM moiety from NAD-capped RNAs (NAD-RNAs). We demonstrate that the deNAMing activity of AbTir (from *Acinetobacter baumannii*) on NAD-RNA specifically produces a cyclic ADPR-RNA, which can be further decapped in vitro by known decapping enzymes. Heterologous expression of the wild-type but not a catalytic mutant AbTir in *E. coli* suppressed cell propagation and reduced the levels of NAD-RNAs from a subset of genes before cellular NAD$^+$ levels are impacted. Collectively, the in vitro and in vivo analyses demonstrate that TIR domain-containing proteins can function as a deNAMing enzyme of NAD-RNAs, raising the possibility of TIR domain proteins acting in gene expression regulation.

Various chemical modifications decorate RNA biomolecules and immensely expand the diversity of the transcriptome[1,2]. RNA modifications can occur internally or terminally[3], and may alter the molecular function, subcellular location, and stability of RNA[4]. Regarding the 5′-terminal RNA structure, besides the most common triphosphate in prokaryotes and the $N^7$-methylguanosine (m$^7$G) cap in eukaryotes, the redox cofactor nicotinamide adenine dinucleotide (NAD$^+$) has emerged as a non-canonical RNA cap structure in diverse organisms. These include prokaryotes such as *Escherichia coli* (*E. coli*)[5–7], *Bacillus subtilis* (*B. subtilis*)[8], *Streptomyces venezuelae* (*S. venezuelae*)[5], and *Mycobacterium smegmatis*[9], and eukaryotes such as yeast[10], mammalian cells[11], and plants[12–15]. Recent studies have provided evidence that NAD-capping of RNA also occurs in Archaea[9,16], establishing the NAD cap

modification as ubiquitous in all three domains of life. Additionally, other non-canonical RNA-caps, such as flavin adenine dinucleotide (FAD), desphospho-coenzyme A (dpCoA), uridine diphosphate glucose (UDP-Glc), uridine diphosphate N-acetyl glucosamine (UDP-GlcNAc), and dinucleotide polyphosphate (Np$_n$N), are also increasingly described[3,17–21].

The identity of NAD-capped RNAs (NAD-RNAs) varies in different species. In bacteria, the protein-coding mRNAs are the major constituents of the NAD-capped transcriptome, and some small regulatory RNAs also tend to carry the NAD cap[6–8,22]. NAD-RNAs in archaea and the eukaryotic kingdom also encompass various types of RNA species[9,16,23], underscoring the importance of studying the biogenesis and function of NAD-RNAs.

[1]State Key Laboratory for Protein and Plant Gene Research, Peking-Tsinghua Joint Center for Life Sciences, School of Life Sciences, Peking University, Beijing 100871, China. [2]Beijing Advanced Center of RNA Biology (BEACON), Peking University, Beijing 100871, China. [3]Department of Botany and Plant Sciences, Institute of Integrative Genome Biology, University of California, Riverside, CA 92521, USA. [4]Institute of Biochemistry, University of Cologne, Cologne 50674, Germany. [5]Max Planck Institute for Plant Breeding Research, Cologne 50829, Germany. [6]Dana-Farber Cancer Institute, Harvard Medical School, Boston, MA 02215, USA. [7]Department of Human Genetics, University of Utah School of Medicine, Salt Lake City, UT 84112, USA. [8]Molecular Medicine Program, Division of Urology, Department of Surgery, University of Utah School of Medicine, Salt Lake City, UT 84112, USA. [9]These authors contributed equally: Xufeng Wang, Dongli Yu. ✉e-mail: xuemei.chen@pku.edu.cn

Thus far, two classes of NAD-RNA decapping enzymes have been identified in prokaryotic or eukaryotic organisms[24] (Supplementary Fig. 1). Class-I decapping enzymes, such as the *E. coli* Nudix hydrolase NudC, cleave the pyrophosphate bond within the NAD cap to liberate the nicotinamide mononucleotide (NMN)[6] (Supplementary Fig. 1). Another bacterial Nudix protein, BsRppH, catalyzes the decapping of NAD-RNAs in Gram-positive *B. subtilis* by releasing NMN[8]. Homologs of NudC in eukaryotic organisms, like Npy1 in yeast[25] and Nudt12/16 in mammalian cells[26], have also been validated as the decapping enzymes. Class-II decapping enzymes, such as the DXO/Rai1 family enzymes in yeast, Arabidopsis, and mammals, remove the entire NAD cap, producing monophosphorylated RNAs (p-RNAs)[11,25,27,28] (Supplementary Fig. 1). The 5′−3′ exoribonucleases Xrn1 and Rat1 in the yeast *Saccharomyces cerevisiae* also serve as NAD cap decapping (deNAD-ding) enzymes, similar to DXO/Rai1, but primarily act on mitochondrial NAD-RNAs[29]. Homologs of DXO/Rai1 have not been found in prokaryotic organisms. More recently, the human glycohydrolase CD38 has been shown to convert NAD-RNAs into ADP-ribose-capped RNAs by liberating nicotinamide (NAM) in vitro[24] (Supplementary Fig. 1), although its in vivo decapping activity still needs validation. It is worth noting that NudC and DXO/Rai1 can act on FAD-capped and dpCoA-capped RNAs as well[30−32] and are thus not specific to NAD-RNAs.

NAD$^+$ can be catabolized by enzymes involved in cellular signaling processes through cleaving the diphosphate bridge joining the two nucleotides or the β-N-glycosidic bond linking ADP-ribose (ADPR) and NAM[33,34]. Recent studies have shown that Toll/interleukin-1 receptor (TIR) domain-containing proteins can act as NAD$^+$-consuming enzymes in bacteria[34−37], mammals[38,39], and plants[38,40,41]. They cleave free NAD$^+$ molecules at the β-N-glycosidic bond into NAM and various forms of ADPR isomers. This TIR domain-mediated NAD$^+$ hydrolase (NADase) activity relies on a conserved catalytic glutamate (Glu/E) residue, which is essential for the activation of downstream immune responses[34,36,38,40,41]. However, it remains unexplored if TIR domain-containing proteins have cleavage activity on the RNA NAD cap.

In this study, we provide evidence revealing that some TIR domain-containing proteins remove the NAM moiety from NAD-RNAs. This TIR domain-mediated deNAMing activity depends on the catalytic Glu/E residue and is enhanced under conditions that promote the oligomerization of TIR domain-containing proteins, such as in the presence of molecular crowding agents or when fused with the tandem Sterile Alpha Motif (tSAM) domain from the human SARM1 protein. We identify the RNA product after AbTir (a TIR-domain containing protein from *Acinetobacter baumannii*) treatment of NAD-RNA as a variant form of cyclic ADPR-RNA (v-cADPR-RNA) that can be further decapped by NudC or DXO/Rai1. TIR domain-containing proteins show decapping activity specific to NAD-RNAs, distinct from other known decapping enzymes. We further demonstrate that AbTir is functional in bacteria – its inducible expression in *E. coli* suppresses growth and significantly decreases the levels of both free NAD$^+$ and NAD-RNAs. SPAAC-NAD-Seq profiling indicates that NAD-RNAs produced from a small subset of genes involved in "molecule transport process" and "oxidoreductase activity" are targeted by AbTir in *E. coli*, further confirming the in vivo deNAMing activity of TIR domains-containing proteins. Finally, we find that an archaeal TIR domain-containing protein, TcpA, also exhibits deNAMing activity on NAD-RNA, extending the deNAMing activity of TIR domains to archaea. Our findings reveal NAD-RNA deNAMing as a molecular function of TIR domain-containing proteins and implicate a role of TIR domain proteins in gene expression regulation.

## Results

### Prokaryotic TIR domain-containing proteins show deNAMing activity on NAD-RNAs

The NADase activity of TIR domain proteins prompted us to test whether they also act on NAD-RNAs. We first expressed a suite of bacterial TIR domain-containing proteins as recombinant proteins in *E. coli*, including AbTir, BtTir, BtpA, TcpC, and TcpF from pathogenic bacteria and TirS and PdTir from non-pathogenic bacteria (Fig. 1a; Supplementary Fig. 2a). All tested TIR domain-containing proteins hydrolyzed free NAD$^+$ to varying degrees in vivo and in vitro (Supplementary Fig. 2b, c), showing NADase activity as reported[35]. Next, to test whether these TIR domain-containing proteins can hydrolyze the RNA NAD cap, we synthesized a 32-nucleotide (nt) NAD-RNA by in vitro transcription using T7 RNA polymerase (Supplementary Fig. 3) and incubated it with the purified TIR domain-containing proteins. The reaction products were subsequently separated by acryloylamino-phenyl boronic acid polyacrylamide gel electrophoresis (APB-PAGE), which can selectively retard the migration of capped RNAs by reacting with the vicinal diol of the cap nucleoside[42]. The results revealed that AbTir from *Acinetobacter baumannii*, BtpA from *Brucella abortus*, and PdTir from *Paracoccus denitrificans*, but not the other tested bacterial TIR domain-containing proteins, showed decapping activity on the NAD-RNA under the assay conditions employed (Fig. 1b). Mutating the putative catalytic Glu/E residues from the TIR domain-containing proteins abolished the activity on both free NAD$^+$ (i.e., NADase activity) and the NAD-RNA (i.e., decapping activity) (Fig. 1c−e).

Time-course assays with AbTir further substantiated the enzymatic decapping activity and showed that AbTir cleaved free NAD$^+$ slightly faster than NAD-RNA in vitro (Fig. 2a, b; Supplementary Fig. 4). As expected, NAD$^+$ depletion accompanying NAM accumulation was detected by high-performance liquid chromatography coupled with mass spectrometry (HPLC-MS) after incubating AbTir with free NAD$^+$ (Fig. 2c). Consistently, NAM was released when AbTir was incubated with an in vitro transcribed NAD-RNA (Fig. 2c), suggesting that AbTir cleaved the β-N-glycosidic bond within the NAD cap to release the NAM moiety (this activity is referred to as deNAMing hereafter; Fig. 2d). NAD-capQ is another method for the quantification of NAD-RNAs among total RNAs by measuring the amount of NAD$^+$ released from RNA after nuclease P1 digestion[43]. To validate the decapping ability of AbTir on native NAD-RNAs, we subjected cellular total RNAs from either Arabidopsis or *E. coli* to AbTir treatment followed by nuclease P1 digestion and NAD-capQ measurement (Fig. 2e). As compared to the catalytically inactive AbTir-E/A, wild-type AbTir caused a significant reduction in the levels of the NAD cap from both in vitro transcribed and native NAD-RNAs (Fig. 2f). Taken together, these results revealed that some bacterial TIR domain-containing proteins can process NAD-RNAs by removing the NAM moiety from the NAD cap, showing a deNAMing activity on in vitro transcribed and native NAD-RNAs.

### Oligomerization enhances the deNAMing activity of AbTir

Next, we evaluated the parameters of the deNAMing assay to optimize the deNAMing reaction. Oligomerization of TIR domains is fundamental for the NADase activity[38,41,44−47]. We first examined the deNAMing activity of AbTir with different concentrations of the purified protein and observed that the deNAMing activity was non-linearly increased with protein concentrations (Supplementary Fig. 5a, b). This observation implied that self-association might also be required for the NAD-RNA deNAMing activity of AbTir, consistent with prior findings that oligomerization of TIR domain proteins enhances the NADase activity on NAD$^+$[47−49]. Polyethylene glycol (PEG), as a molecular crowding agent, can simulate a crowded environment inside cells, and promote the self-association of TIR domains[38,41]. We thus applied different types of PEGs in the reaction and found that the deNAMing activity was improved in the presence of certain PEGs (Supplementary Fig. 5c). The same amount of NAD-RNAs was cleaved by AbTir in 2 h in the presence of PEG6000 as that in 16 h in the absence of PEG6000 (Supplementary Fig. 5d). The tandem Sterile Alpha Motif (tSAM) domain of human SARM1 (Sterile alpha and HEAT/Armadillo motif) enhances the NADase activity of TIR domains by facilitating their

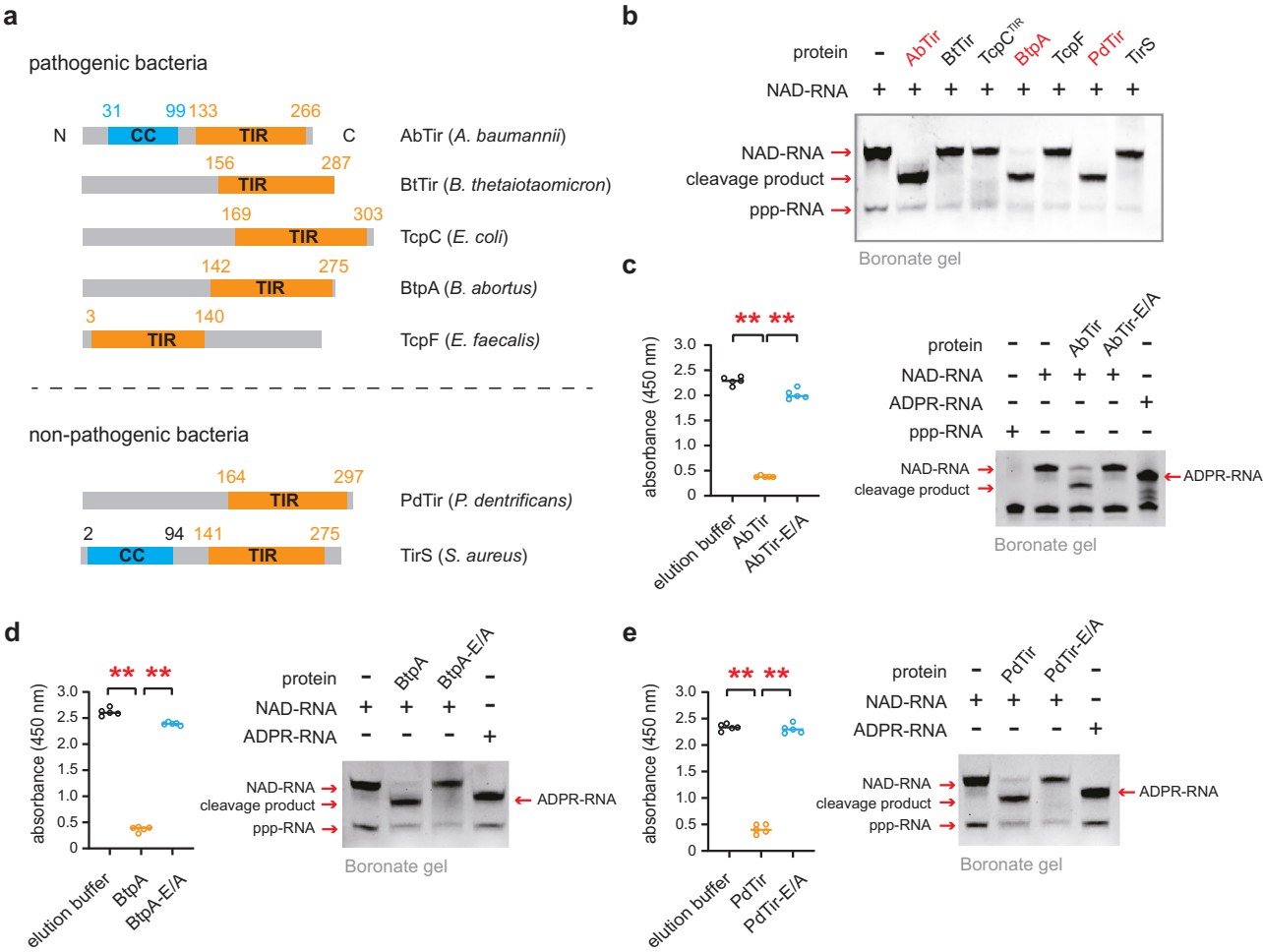

**Fig. 1 | Bacterial TIR domain-containing proteins cleave the NAD cap of NAD-RNAs. a** Diagrams of the tested bacterial TIR domain-containing proteins. The numbers above indicate the start and end amino acids of the annotated domains. CC, coiled coil; TIR, Toll/interleukin-1 receptor. **b** An acryloylaminophenyl boronic acid (APB) gel showing the outcome of decapping assays, in which various bacterial TIR domain-containing proteins were incubated with an in vitro transcribed NAD-RNA. Three TIR domain-containing proteins (AbTir, BtpA, and PdTir) showed activities toward the NAD-RNA. A reaction without purified protein added was included for comparison. The positions of NAD-RNA, ppp-RNA and the cleavage product are indicated. Assays for the enzymatic activities of AbTir (**c**), BtpA (**d**), and PdTir (**e**) and their corresponding catalytic mutants. Both the NADase activities on free NAD$^+$ (left) and the decapping activities on the NAD-RNA (right) were dependent on the putative catalytic Glu/E residues. The relative NAD$^+$ content after the NADase assay was measured with the NAD/NADH Quantitation Kit by monitoring the absorbance values at 450 nm. Horizontal bars represent median values of five independent experiments ($n = 5$); [**] $P \leq 0.01$ (calculated by the non-parametric Mann-Whitney U-test, two-sided). In vitro transcribed ppp-RNA and ADPR-RNA were used as markers for comparison to the NAD-RNA cleavage products. Exact $P$-values and source data are provided as a Source Data file.

multimerization[39,50,51]. We fused this tSAM domain (hSARM1$^{tSAM}$; residues 409-561aa) to AbTir$^{TIR}$ (TIR domain of AbTir, residues 133-266aa) (Supplementary Fig. 5e) and purified the fusion protein hSARM1$^{tSAM}$-AbTir$^{TIR}$ from *E. coli* (Supplementary Fig. 5f). As expected, fusion of hSARM1$^{tSAM}$ to AbTir$^{TIR}$ significantly enhanced the deNAMing activity of AbTir (Supplementary Fig. 5g). Collectively, these results indicated that self-association of AbTir plays a pivotal role in bolstering its deNAMing activity on NAD-RNAs.

### Identification of the RNA product of deNAMing by AbTir

The RNA product after AbTir cleavage showed faster migration on APB-PAGE than in vitro transcribed ADPR-RNA (Fig. 1c), which prompted us to investigate the identity of this cleavage product. As a NADase enzyme, AbTir cleaves free NAD$^+$ into NAM and a cADPR isomer known as v-cADPR or 2'-cADPR, meaning that AbTir has a cyclase activity in addition to the cleavage activity[35,37,47]. Thus, we assume that the RNA product of AbTir might possess a cyclic ADPR cap (Fig. 3a). A recent study demonstrated that a conserved tryptophan of AbTir is required for the cyclization of ADPR after the hydrolysis reaction with

free NAD$^{+47}$. Mutating this residue from tryptophan (Trp/W) to alanine (Ala/A) significantly reduced, but did not abolish, the production of 2'-cADPR, compared to wild-type AbTir[47]. We thus hypothesized that mutating the tryptophan residue will result in the generation of non-cyclic ADPR-RNA. We generated the AbTir-W204A mutant and performed the NADase and decapping assays with the purified protein (Fig. 3b, c). Consistent with the previous report[47], a significant reduction in the NADase activity was observed (Fig. 3b). The decapping activity was also reduced, as reflected by the input NAD-RNA being incompletely consumed by AbTir-W204A (Fig. 3c). Interestingly, two RNA products were produced by AbTir-W204A, with one showing the same migration as the AbTir product and the other showing the same migration as the in vitro transcribed ADPR-RNA (Fig. 3c), suggesting that this tryptophan residue plays an important role in determining the nature of the AbTir NAD-RNA deNAMing product. This observation also implied that the RNA product from AbTir treatment should be a certain type of cyclic ADPR-RNA. Unfortunately, we failed to obtain cADPR-RNA by in vitro transcription using cADPR in place of ATP and v-cADPR is not commercially available for use in in vitro transcription,

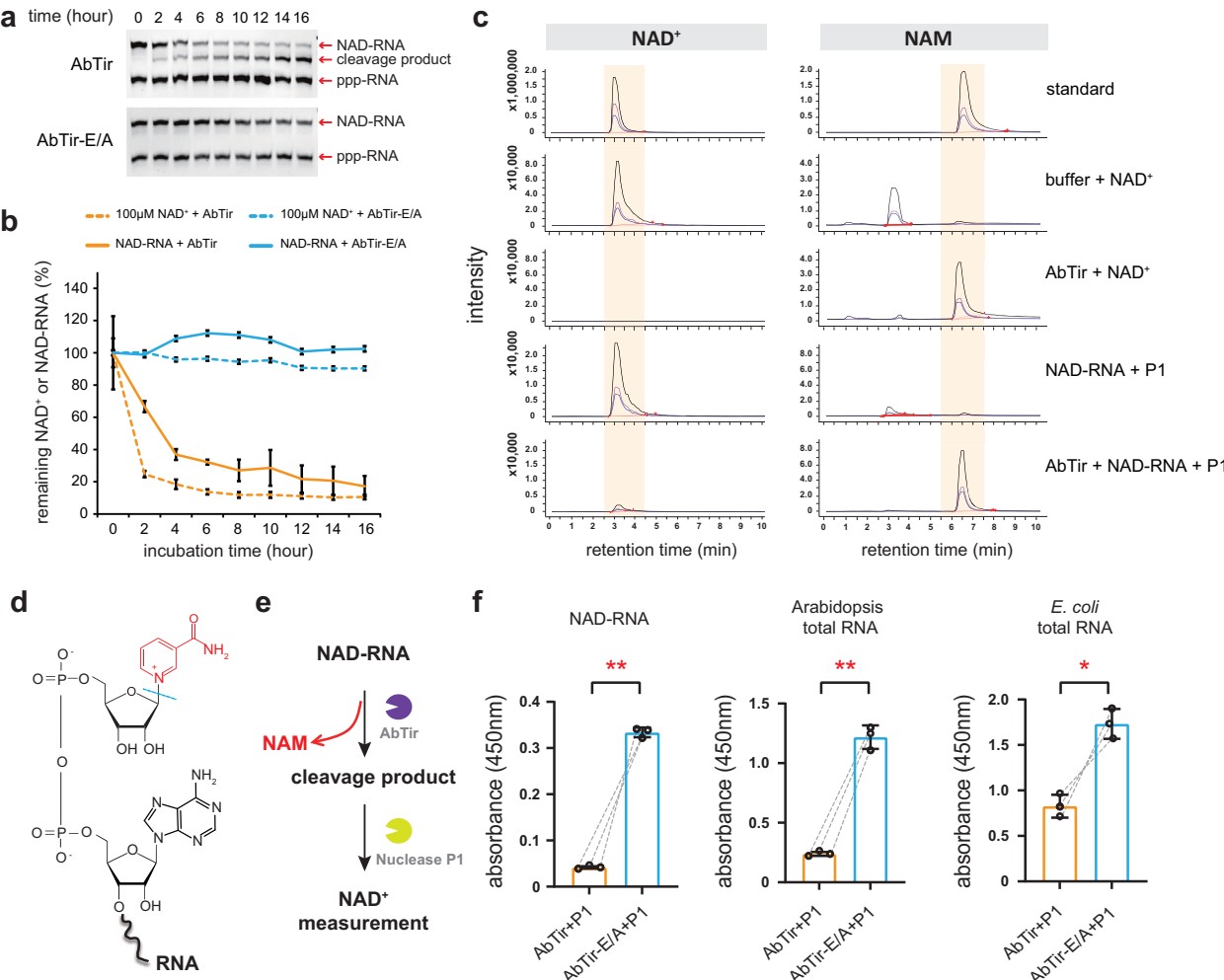

**Fig. 2 | AbTir exhibits deNAMing activity on NAD-RNAs. a** A time course of NAD-RNA decapping by wild-type AbTir and its catalytic mutant AbTir-E/A. An in vitro transcribed NAD-RNA was incubated with AbTir (top panel) or its catalytic mutant AbTir-E/A (bottom panel) for the indicated duration and RNAs in the reaction were resolved in denaturing APB gels. **b** Comparison of the reaction kinetics of the NADase activity on free NAD$^+$ and the decapping acticity on NAD-RNA by AbTir. The catalytic mutant AbTir-E/A was included as a negative control. For each time point, the remaining substrates were analyzed from three independent experiments with error bars representing mean ± SD ($n = 3$, Supplementary Fig. 4). **c** HPLC-MS analysis demonstrating that free NAD$^+$ or the NAD cap from the NAD-RNA was largely destroyed, accompanied by nicotinamide (NAM) accumulation after AbTir treatment. For the detection of NAD$^+$ in RNA, the RNA was treated with nuclease P1 (P1) before HPLC-MS analysis. The three product ions from NAD$^+$ ([M + H]$^+$ 664.00 > 136.00, 664.00 > 427.90, 664.00 > 523.95) and NAM ([M + H]$^+$

123.00 > 80.00, 123.00 > 78.00, 123·00 > 53.00) are indicated by black, pink and blue lines, respectively. **d** Diagram of NAD-RNA and its deNAMing cleavage site by AbTir. The cleavage site in the NAD cap is marked with a blue dashed line, and the NAM structure is highlighted in red color. **e** The experimental pipeline showing NAD cap breakdown from NAD-RNA by pretreatment with AbTir. After AbTir cleavage, the NAD cap should be destroyed and should not be detected after nuclease P1 digestion of the RNA. **f** Measurement of NAD cap content using the pipeline in (**e**) after AbTir (orange) or AbTir-E/A (light blue) treatment. An in vitro transcribed NAD-RNA (0.5 μg) was used as a positive control. 500 μg total RNA from Arabidopsis or E. coli was used to validate the deNAMing activity of AbTir. Error bars represent mean ± SD, which were calculated from three biologically independent samples ($n = 3$); [**] $P ≤ 0.01$; [*] $P ≤ 0.05$ (calculated by the Student's $t$-test, two-sided). Exact $P$-values and source data are provided as a Source Data file.

which impeded the ability to narrow down the identity of the RNA product by APB-PAGE.

It is known that adenosine diphosphate-ribosyl cyclase (ADPRC) purified from *Aplysia californica* is a bifunctional enzyme that releases NAM from NAD$^+$ and cyclizes ADPR to cADPR[52]. Another bifunctional enzyme, human CD38, catalyzes the hydrolysis of NAD$^+$ as a glycohydrolase, but the cyclase activity is weak, resulting in 97% of the products being ADPR[53–55]. Both ADPRC and CD38 are able to remove the NAM moiety from in vitro transcribed NAD-RNA[6,24]. The RNA product of ADPRC-catalyzed NAD-RNA deNAMing was not determined. While Abele et al. reported ADPR-RNA as one of the cleavage products of CD38 on in vitro transcribed NAD-RNA[24], they did not explore other potential products, such as cADPR-RNA and v-cADPR-RNA. We analyzed the reaction products of AbTir, ADPRC, and CD38 on free NAD$^+$

by HPLC-MS, and found that v-cADPR[47,56] and ADPR were detected for AbTir and CD38, respectively (Supplementary Fig. 6), as reported[47,53,54,56]. However, a clear peak corresponding to the ADPR standard was detected for ADPRC (Supplementary Fig. 6), not cADPR as previously reported[52]. After knowing the products produced by these enzymes on free NAD$^+$, we compared the products of AbTir, ADPRC and CD38 on NAD-RNA. In APB-PAGE, the RNA products of ADPRC and CD38 showed the same migration as the in vitro transcribed ADPR-RNA (Fig. 3d). To unambiguously determine the identities of the RNA products after ADPRC and CD38 cleavage, we established an HPLC-MS analysis pipeline. The in vitro transcribed NAD-RNA was first incubated with either ADPRC or CD38 to release the NAM moiety from the NAD-RNA, and then the RNA products were further digested by nuclease P1 into single nucleotides, which were

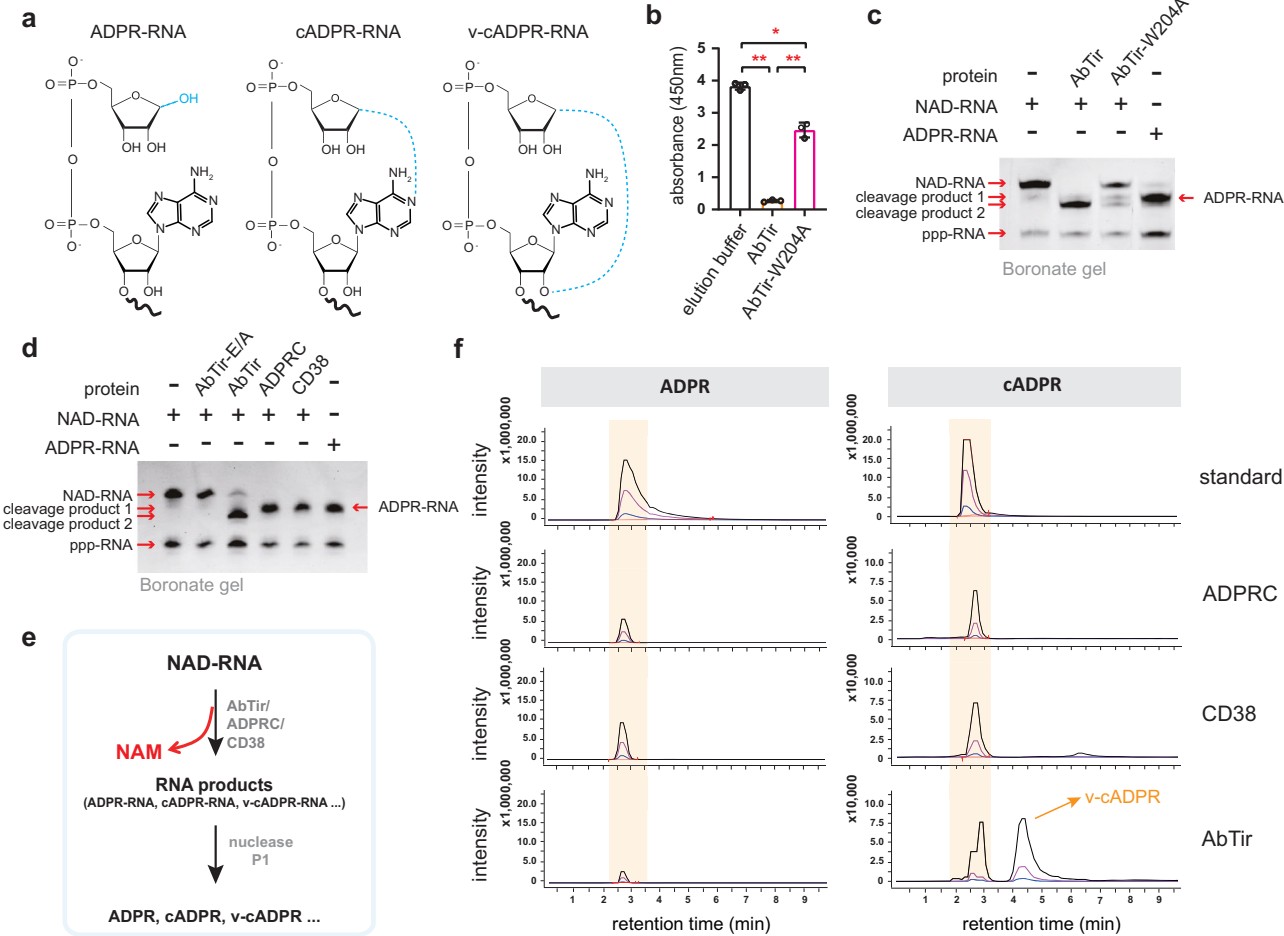

**Fig. 3 | Identification of the RNA product of NAD-RNA deNAMing by AbTir.**
**a** Diagrams showing the possible products (ADPR-RNA, cADPR-RNA, and v-cADPR-RNA) of NAD-RNA deNAMing by TIR domain-containing proteins. **b** NADase assays (see Methods) to measure the hydrolytic activity of wild-type AbTir and its W204A mutant on free NAD⁺. The W204A mutation significantly reduced the NADase activity. Error bars represent mean ± SD, which was calculated from three independent experiments ($n = 3$); [**] $P \leq 0.01$ (calculated by the Student's $t$-test, two-sided). **c** NAD-RNA decapping assays for wild-type AbTir and its W204A mutant. The AbTir-W204A mutant showed reduced deNAMing activity compared with wild-type AbTir but generated a new cleavage product corresponding to the in vitro transcribed ADPR-RNA. **d** An APB gel showing the products of NAD-RNA decapping by AbTir, ADPRC and CD38. An in vitro transcribed NAD-RNA was used as the substrate. ADPRC and CD38 converted the NAD-RNA to a product of the same mobility as the in vitro transcribed ADPR-RNA. The RNA product from AbTir had higher mobility in the APB gel than the products from ADPRC or CD38, or the in vitro transcribed ADPR-RNA. **e** A pipeline used for identifying the deNAMing products by AbTir, ADPRC, and CD38. **f** HPLC-MS analyses following the pipeline in (**e**) demonstrated that both ADPR-RNA and cADPR-RNA were the products generated by ADPRC and CD38, while v-cADPR-RNA was produced by AbTir. Commercial ADPR and cADPR were used as standards for the HPLC-MS analysis. The three product ions of cADPR ([M + H]⁺ 541.80 > 136.15, 541.80 > 427.90, 541.80 > 347.90) and ADPR ([M + H]⁺ 559.80 > 136.05, 559.80 > 347.95, 559.80 > 427.95) are indicated by black, pink and blue lines, respectively. Exact $P$-values and source data are provided as a Source Data file.

subjected to HPLC-MS analysis (Fig. 3e). This analysis detected both ADPR and cADPR for reactions with CD38 and ADPRC (Fig. 3f), meaning that both ADPR-RNA and cADPR-RNA are the RNA products upon CD38 or ADPRC treatment. With this pipeline (Fig. 3e), we found that the same AbTir NADase product, i.e., v-cADPR, was detected in the AbTir deNAMing reaction on the NAD-RNA (Fig. 3f; Supplementary Fig. 6), indicating that v-cADPR-RNA was the specific RNA product of AbTir-mediated NAD-RNA deNAMing.

## AbTir specifically cleaves the RNA NAD cap in vitro as an initial step for decapping

Various moieties, such as the triphosphate group, NAD⁺, dpCoA, FAD, and NpₙN, have been reported to be covalently attached to the 5′ terminus of RNA and might affect RNA stability[3,17–19]. In vitro and in vivo, NAD-capped RNAs can be decapped by Nudix and DXO/Rai1 family enzymes, originally identified in disparate prokaryotic and eukaryotic organisms[6,11,24]. Both classes of enzymes decap NAD-RNAs but also decap RNAs with other non-canonical caps, such as FAD and

dpCoA[30,31,57,58]. The primary product of decapping is p-RNA, which can undergo rapid RNA decay.

To examine the substrate specificity of AbTir, we used a set of in vitro transcribed RNAs with non-canonical caps (Supplementary Fig. 3). We found that NudC and DXO can indiscriminately cleave the caps of NAD-RNA, FAD-RNA, dpCoA-RNA, and ADPR-RNA (Fig. 4a). NudC also showed robust cleavage activity on Ap₄A-RNA, while DXO did not. Among all the substrate RNAs tested, AbTir only acted on NAD-RNA (Fig. 4a).

Because in vitro transcribed ADPR-RNA can be decapped by NudC and DXO, we tested whether the v-cADPR-RNA generated by AbTir can be further decapped by NudC or DXO. We added NudC and DXO directly to the reaction after the NAD-RNA was incubated with AbTir. We observed that the v-cADPR-RNA was further cleaved into p-RNA by NudC or DXO (Fig. 4b). Therefore, TIR domain-containing proteins can specifically remove NAM from the NAD cap, and a second decapping step is required to produce p-RNA (Fig. 4c), suggesting that these enzymes might work together to specifically decap NAD-RNAs. A

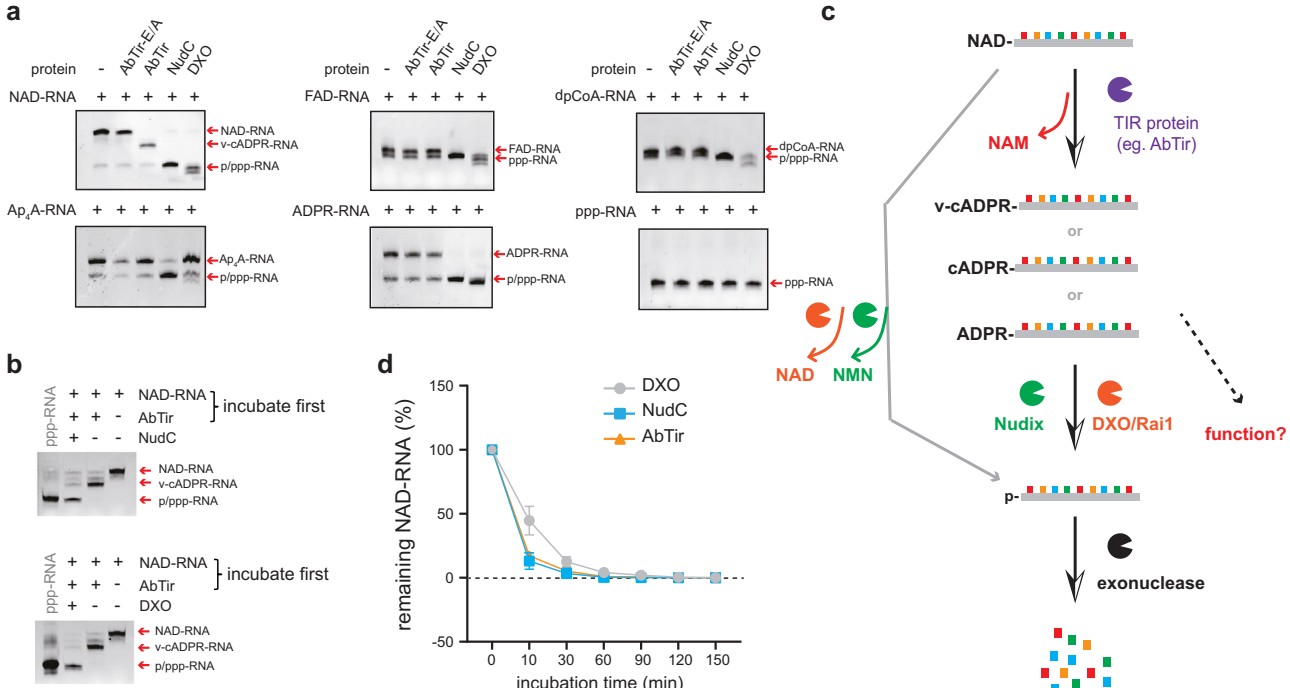

**Fig. 4 | Specificity of AbTir on non-canonical RNA caps in comparison to NudC and DXO. a** APB gels showing in vitro transcribed RNAs with various non-canonical caps (NAD, FAD, dpCoA, Ap$_4$A, ADPR, and triphosphate) and decapping activities of AbTir, NudC, and DXO on these RNAs. Reactions without the enzymes were included for comparison. The red arrows represent the positions of the indicated RNA species in APB gels. **b** APB gels showing the combined effects of AbTir and NudC/DXO enzymes. ppp-RNA was loaded as a marker to show the position of the final cleavage product. NAD-RNA was incubated first with AbTir for the deNAMing assay, and then NudC or DXO was added into the reaction for further incubation. **c** Schematic of NAD-RNA hydrolysis by Nudix family enzymes, DXO/Rai1 deNAD-ding enzymes, and TIR-domain deNAMing enzymes. **d** Rates of NAD-RNA decapping by AbTir, NudC, and DXO. Data points represent mean ± SD. For each time point, the reaction kinetics was analyzed from three independent decapping assays (Supplementary Fig. 7). The y-axis indicates the proportions of remaining NAD-RNAs after decapping reactions. In each reaction, 500 ng NAD-RNA was incubated with 10 μM AbTir/NudC/DXO. Source data are provided as a Source Data file.

recent report showed that ADPR-RNA exists in archaea and can be generated from NAD-RNA through the spontaneous loss of NAM[16]. Our findings suggest that TIR-catalyzed NAM removal is another mechanism to generate RNAs capped by ADPR or its variants (Fig. 4c).

We compared the rate of NAD-RNA decapping by these three enzymes through a time-course assay. The results showed that AbTir could hydrolyze the NAD cap at a similar rate as NudC and DXO (Fig. 4d; Supplementary Fig. 7). These results suggest that multiple mechanisms might exist for NAD-RNA decapping in vivo. In addition to Nudix hydrolases and DXO/Rai1 family enzymes, TIR domain-containing proteins may also play a role in removing the NAD cap.

## AbTir deNAMs native NAD-RNAs
Having shown that AbTir removes NAM from in vitro transcribed NAD-RNAs, we sought to determine whether AbTir deNAMs cellular NAD-RNAs. In previous studies, it was shown that NAD-RNAs can be selectively biotinylated by Copper-catalyzed azide-alkyne cycloaddition or strain-promoted azide-alkyne cycloaddition (SPAAC) click chemistry and biotinylated RNAs can be detected by gel blots or captured for high throughput sequencing[6,7,15]. Unlike NAD-RNA, we found that ADPR-RNA was not biotinylated by the SPAAC reaction (Supplementary Fig. 8). Based on this observation, we envisioned that NAD-RNAs could not be biotinylated if pre-treated with AbTir (Fig. 5a). To test this, we treated an in vitro transcribed NAD-RNA and native mRNAs isolated from Arabidopsis and *E. coli* with wild-type AbTir or catalytically inactive AbTir-E/A, and then performed the SPAAC reaction followed by gel analysis. Treatment with wild-type AbTir significantly reduced the detection of biotin-linked RNAs, while treatment with AbTir-E/A had no effect on signal detection (Fig. 5b, c). This showed that AbTir can deNAM Arabidopsis or *E. coli* NAD-RNAs in vitro.

To identify the NAD-RNA species acted upon by AbTir, we performed SPAAC-NAD-Seq[15] using *E. coli* ribosomal RNA (rRNA)-depleted RNA after AbTir or AbTir-E/A treatment. The results of SPAAC-NAD-Seq were highly reproducible among biological replicates (Supplementary Fig. 9). After AbTir treatment, we identified only 12 NAD-RNA-producing genes with *E. coli* rRNA-depleted RNA, whereas after treatment with AbTir-E/A, we identified 1,057 NAD-RNA-producing genes (Ratio of NAD-RNA-Seq/regular RNA-Seq ≥ 2 & FDR ≤ 0.05; Fig. 5d; Supplementary Data 1 and 2), suggesting that almost all NAD-RNAs are sensitive to deNAMing activity by AbTir in vitro.

In a previous report using NAD tagseq II, a total of 279 NAD-RNA-producing genes were identified in *E. coli*[7]. Among these genes, 248 had expressed homologs in the BL21 (DE3) strain used in our study, and 165 (~67%) of them were also identified as NAD-RNA-producing genes in our SPAAC-NAD-Seq analysis (Supplementary Fig. 10a). This demonstrates the accuracy and robustness of our SPAAC-NAD-Seq profiling. The increased number of NAD-RNA-producing genes identified in our analysis may be attributed to the limited sensitivity of Oxford nanopore sequencing used in NAD tagSeq II, as well as differential expression of cognate genes under different conditions. The newly identified 1,057 NAD-RNA-producing genes are predominantly protein-coding genes, with a few from non-coding RNAs (Supplementary Fig. 10b). These NAD-RNA-producing genes tend to be shorter and have lower expression levels (Supplementary Fig. 10c, d). Gene ontology (GO) analysis revealed enrichment in functional categories related to "RNA metabolic process" and "regulation of gene expression" (Supplementary Fig. 10e). We specifically focused on the top 20 NAD-RNAs identified in the NAD tagSeq II assay and found that 15 (75%) of them were highly enriched as NAD-RNAs after AbTir-E/A treatment (Fig. 5f). Among the remaining 5 NAD-RNAs, four of them (*sibD*, *amyA*,

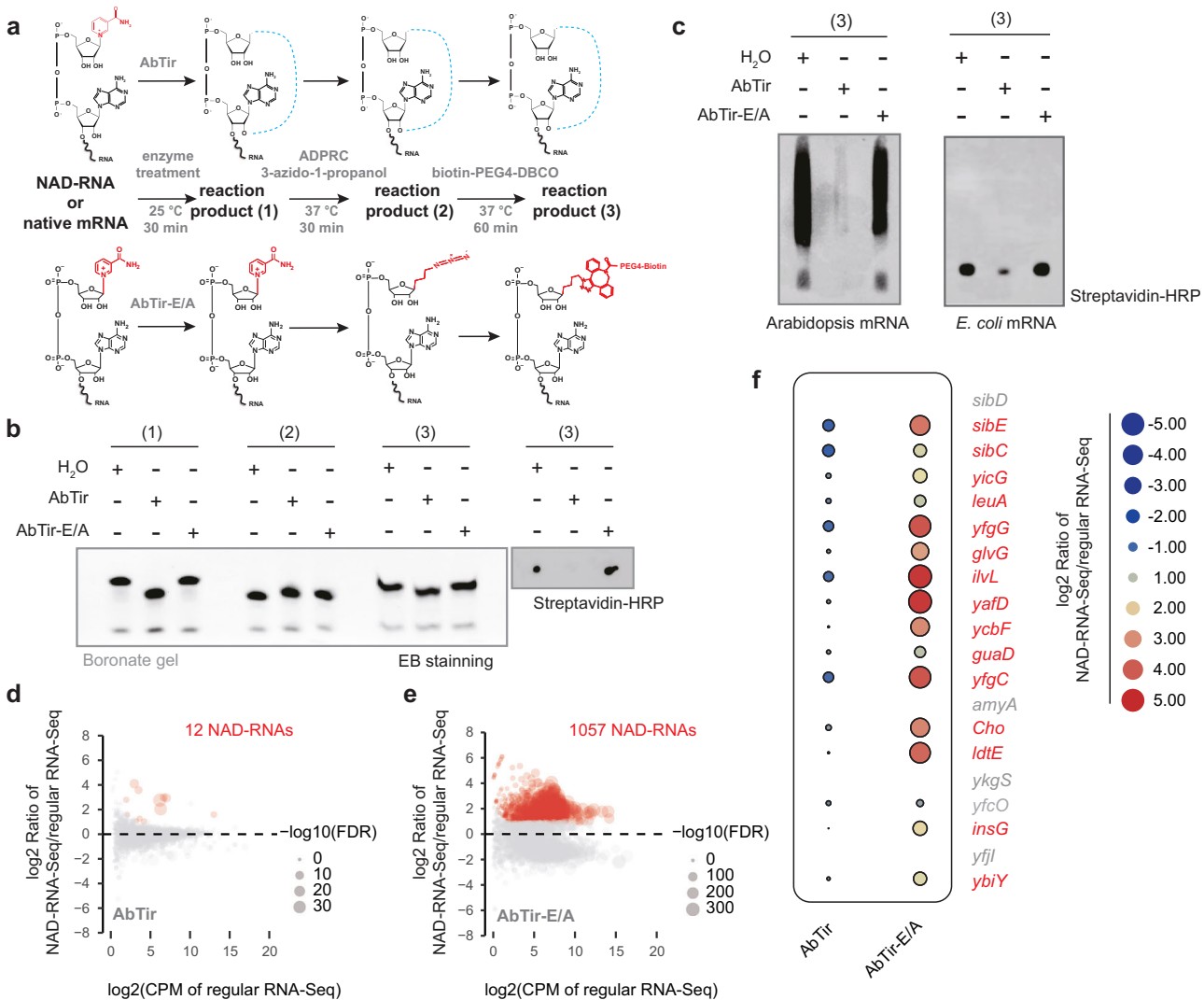

**Fig. 5 | AbTir deNAMs native NAD-RNAs. a** Workflow to determine the deNAMing activity of AbTir on cellular NAD-RNAs. In vitro transcribed NAD-RNA or total cellular RNAs were pre-treated with AbTir or AbTir-E/A and then subjected to two steps (ADPRC and SPAAC reactions) that convert NAD-RNAs to biotinylated RNAs. In the presence of AbTir, NAD-RNAs are deNAMed and the subsequent biotinylation cannot occur. In the absence of AbTir, the ADPRC reaction replaces the nicotinamide of the NAD cap with 3-azido-1-propanol, which is then linked to biotin-PEG4 in the SPAAC reaction. **b** An in vitro transcribed NAD-RNA was used to test the pipeline in (**a**). The numbers in brackets stand for the reaction products at different steps from the pipeline in (**a**). An APB gel was used to resolve the different products, which were visualized by ethidium bromide (EB) staining. To detect biotinylated RNAs, a dot blot was performed using nylon N+ membrane and biotin signals were detected with streptavidin-HRP. **c** Gel blots showing biotinylated RNAs after Arabidopsis and *E. coli* RNAs went through the pipeline in (**a**). One microgram Arabidopsis mRNA or 1 μg *E. coli* rRNA-depleted RNA was used for each assay. The RNA products were resolved in 2% agarose gels and transferred to nylon N+ membranes for the detection of biotin signals with streptavidin-HRP. **d**, **e** Scatter plots showing NAD-RNAs identified by SPAAC-NAD-Seq in *E. coli* RNA after AbTir (**d**) or AbTir-E/A (**e**) pretreatment. The log2 ratio of CPM (Counts Per Million) between NAD-RNA-Seq and regular RNA-Seq is plotted against the log2 CPM of regular RNA-Seq. The carmine dots with different sizes represent enriched NAD-RNA-producing genes, which was identified based on the criteria of "NAD-RNA-Seq/regular RNA-Seq ≥ 2 and FDR ≤ 0.05". **f** Heatmap showing the levels of 20 known NAD-RNAs after AbTir pretreatment. Fifteen (labeled with red fonts) of the 20 known NAD-RNAs were identified as NAD-RNAs in our study, all of which were decapped by AbTir pretreatment. Four genes (*sibD*, *amyA*, *ykgS*, and *yfjI*) in gray fonts did not have cognate genes in the *E. coli* BL21 (DE3) strain and another gene *yfcO* was not detected as an NAD-RNA-producing gene. Source data are provided as a Source Data file.

*ykgS*, and *yfjI*) did not have cognate genes in the *E. coli* BL21 (DE3) strain and one of them (*yfcO*) was not detected as an NAD-RNA. However, after treatment with AbTir, the enrichment of these NAD-RNAs was significantly reduced (Fig. 5f; Supplementary Fig. 11). This indicates the strong decapping activity of AbTir on native RNA.

### AbTir targets a subset of NAD-RNAs in *E. coli*

To determine the in vivo decapping activity of AbTir, we expressed the wild-type AbTir or the catalytically inactive AbTir-E/A (Fig. 6a) in the *E. coli* BL21 (DE3) strain. We monitored cell proliferation by measuring the OD values at 600 nm (OD600) after AbTir induction with different

concentrations of IPTG. *E. coli* cells expressing AbTir showed decreased cell proliferation compared to those expressing AbTir-E/A (Fig. 6b; Supplementary Fig. 12a–c). This decrease in cell proliferation may be due to the consumption of NAD+ in *E. coli* cells by AbTir, as previously reported[35,47] and confirmed in our measurements (Fig. 6c; Supplementary Fig. 12d). We sought to determine whether AbTir also caused NAD-RNA deNAMing in bacteria. To this end, we first measured the endogenous NAD+ content at different time points after IPTG addition (Fig. 6d; Supplementary Fig. 13), and found a time point, at which AbTir accumulated but levels of free NAD+ were not affected (Fig. 6d, e). At this time point (15 min after induction with 0.1 mM

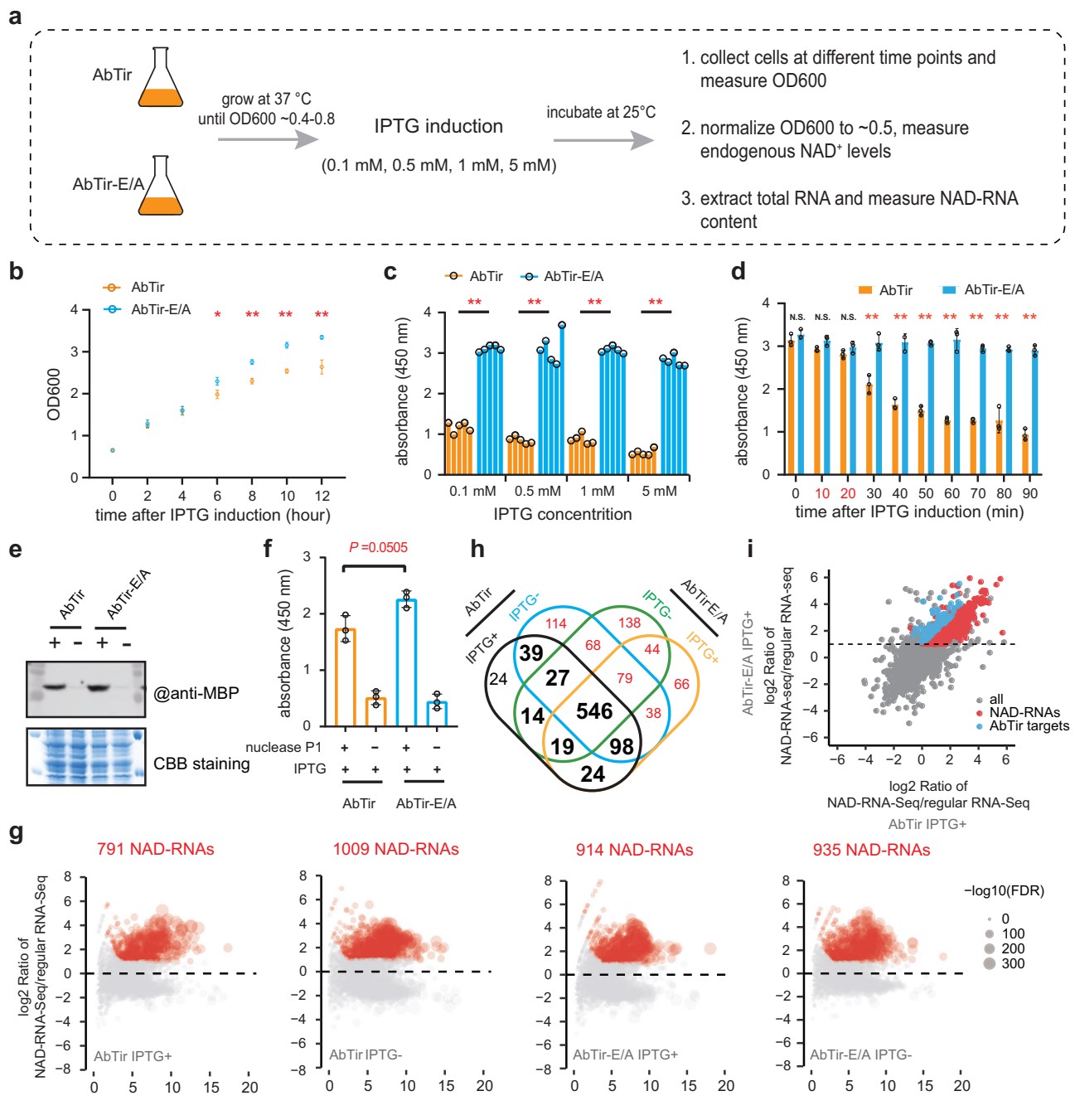

**Fig. 6 | AbTir is a deNAMing enzyme in *E. coli* cells. a** Diagram showing heterologously expressing AbTir or AbTir-E/A in *E. coli* followed by evaluation of its effects on cell growth, endogenous NAD⁺, and NAD-RNA levels. **b** Measurement of OD600 values at the indicated time points after the addition with 1 mM IPTG. Three biological replicates were measured at each time point. **c** Levels of NAD⁺ in *E. coli* cells expressing AbTir (orange) or AbTir-E/A (light blue) at 2 h after induction. Each bar stands for a biological replicate. **d** Levels of NAD⁺ in *E. coli* expressing AbTir (orange) or AbTir-E/A (light blue) after induction with 0.1 mM IPTG. Three biological replicates were measured at each time point. **e** Western blot to detect MBP-tagged AbTir and AbTir-E/A proteins with anti-MBP, demonstrating that the recombinant proteins were present at 15 min after IPTG induction. CBB, Coomassie Brilliant Blue. **f** Quantification of NAD cap content in total RNA isolated from *E. coli* cells expressing AbTir or AbTir-E/A after 15 min IPTG induction. Nuclease P1 (+) was used to digest the total RNA to release the NAD cap. Samples treated with DEPC water (-) were used as negative controls. **g** Scatter plots showing NAD-RNAs identified by SPAAC-NAD-Seq in *E. coli* expressing AbTir or AbTir-E/A with or without

IPTG induction. The log2 ratio of CPM between NAD-RNA-Seq and regular RNA-Seq is plotted against the log₂(CPM) of regular RNA-Seq. The labeled dots (carmine color) represent the identified NAD-RNAs. **h** Venn diagram showing the overlap of NAD-RNAs identified in *E. coli* expressing AbTir or AbTir-E/A with or without IPTG induction. Bold numbers represent NAD-RNAs identified in *E. coli* expressing AbTir that were also identified in the other three datasets. Red numbers represent the NAD-RNA-producing genes that were specifically identified in *E. coli* without AbTir expression. **i** Comparison of NAD-RNA levels between *E. coli* expressing AbTir and AbTir-E/A. NAD-RNA levels are defined as the log2 ratio of NAD-RNA-Seq/regular RNA-Seq. Red points represent NAD-RNAs identified in *E. coli* expressing AbTir-E/A. Blue points represent the NAD-RNAs that have significantly lower enrichment levels in *E. coli* cells expressing AbTir than in *E. coli* expressing AbTir-E/A. For (**b**, **d**, **f**), average values from three biological replicates are shown with error bars denoting mean ± SD. [**] *P* ≤ 0.01; [*] *P* ≤ 0.05; N.S., not significant (calculated by the nonparametric Mann-Whitney U-test, two-sided). Exact *P*-values and source data are provided as a Source Data file.

IPTG), we collected AbTir- and AbTir-E/A-expressing cells and isolated total RNAs. After removing free NAD$^+$, we performed nuclease P1 digestion on the total RNA and quantified the NAD cap content using the NAD-capQ assay. We observed a significant, albeit slight, reduction in NAD cap content in total RNA from *E. coli* cells expressing AbTir compared to AbTir-E/A (Fig. 6f). This suggests a global reduction in NAD-RNA levels in vivo after AbTir expression.

To determine changes in specific NAD-RNA transcripts, SPAAC-NAD-Seq[7,15] was carried out with the same total RNAs from *E. coli* cells expressing AbTir and AbTir-E/A (n = 3 for each group). After rRNA depletion, the retained RNA was subjected to treatments that converted NAD-RNAs to biotin-RNAs for sequencing (referred to as "NAD-RNA-Seq"). A portion of the rRNA-depleted RNA was used for direct RNA sequencing (referred to as "regular RNA-Seq") as a control. The sequencing libraries were of high reproducibility, according to clustering analysis (Supplementary Fig. 14). We identified a total of 791 NAD-RNA-producing genes in *E. coli* cells expressing AbTir after IPTG induction, which is fewer than the number identified in *E. coli* cells without IPTG induction or in *E. coli* cells expressing AbTir-E/A (Fig. 6g; Supplementary Data 3–6). Overlap analysis revealed that 767 (~97%) of the NAD-RNA-producing genes identified in *E. coli* cells expressing AbTir were consistently identified in the other three datasets, demonstrating the accuracy and effectiveness of SPAAC-NAD-Seq (Fig. 6h). A total of 547 NAD-RNA-producing genes were specifically identified in *E. coli* cells without AbTir expression (Fig. 6h), suggesting that they are the possible decapping targets of AbTir in vivo. Using more stringent criterion of at least a 2-fold change and $P \leq 0.05$, we identified 78 NAD-RNA-producing genes that had higher levels of NAD-RNAs in *E. coli* cells expressing AbTir-E/A than in cells expressing AbTir (Fig. 6i; Supplementary Fig. 15; Supplementary Data 5). These genes are considered deNAMing targets of AbTir in vivo. Functional enrichment analysis showed that these AbTir-sensitive NAD-RNA-producing genes are most strongly enriched in GO terms including "molecule transport process" and "oxidoreductase activity". These findings indicate that AbTir, as a deNAMing enzyme, preferentially targets a subset of NAD-RNAs in *E. coli* cells. However, it should be noted that the number of AbTir deNAMing targets is likely underestimated because we collected cells for RNA isolation only at 15 min after IPTG induction in order to avoid the disturbance of free NAD$^+$.

## TcpA, a TIR domain-containing protein from archaea, also exhibits deNAMing activity

Recent studies have demonstrated the presence of NAD-RNAs in archaeal model organisms, with the highest NAD cap concentration reported so far[9,16], thereby extending the existence of NAD-RNAs to all three domains of life. Given the widespread occurrence of NAD-RNAs and the conservation of TIR domains across various phyla[59,60], we hypothesize that the deNAMing capacity of TIR domain might be evolutionarily conserved. To explore this possibility, we investigated the deNAMing ability of several TIR homologs from archaea to plants and humans (Fig. 7a). We first expressed two TIR domain-containing proteins, TcpA and TcpO from archaeal species *Theionarchaea archaeon* and *Methanobrevibacter olleyae*, respectively, in *E. coli*. Strikingly, TcpA depleted free NAD$^+$ as expected[35] and cleaved the RNA NAD cap, similar to AbTir (Fig. 7b, c). Both activities were abolished in the putative catalytic mutant (TcpA-E/A) (Fig. 7b, c). However, we did not observe any decapping activity for TcpO, despite its apparent NADase hydrolytic activity[35] (Fig. 7b, c), suggesting that not all TIR domain-containing proteins possess the NAD-RNA deNAMing activity.

Additionally, TIR domains are present in a wide range of microbial genomes[34,36], and certain microbial TIR-domain NADases have been co-opted as virulence factors[34,37,61]. For instance, HopAM1, a TIR domain-containing effector protein encoded by the plant pathogen *Pseudomonas syringae* DC3000, has been shown recently to suppress plant immunity by hydrolyzing NAD$^+$ and producing a novel cADPR variant

(v2-cADPR or 3′-cADPR) in plant cells[34,37]. With the purified protein (Fig. 7d), we confirmed the NADase activity of HopAM1, which was dependent on the putative catalytic Glu/E residue (E191) (Fig. 7e). However, we did not observe the decapping activity of HopAM1 on NAD-RNAs, even with high concentrations of the HopAM1 protein applied (Fig. 7f). Given the cyclization site of 3′-cADPR, it is theoretically impossible to generate a 3′-cADPR-RNA, because the 3′-OH on the ribose of the cap adenosine is occupied by the RNA body. It is still unknown if there are other TIR domain-containing effectors having decapping ability on NAD-RNAs, which might enzymatically subvert plant immune systems.

The classical human TIR NADase SARM1, which can hydrolyze free NAD$^+$ efficiently[39,51] (Supplementary Fig. 16a, b), was also examined and proven incapable of cleaving the RNA NAD cap (Supplementary Fig. 16c). To further extend our testing to plants, we selected several TIR domain proteins, including BdTIR from *Brachypodium distachyon*, OsTIR from *Oryza sativa*, the TIR domain of L7 from *Linum usitatissimum*, and the TIR domains of RPS4 (*resistance to Pseudomonas syringae 4*) and RBA1 (Response to HopBA1) from *Arabidopsis thaliana*, for expression in *E. coli*. However, consistent with previous reports, the plant TIR domains did not deplete free NAD$^+$ in our in vitro NADase assay due to their weak NADase activities. Moreover, some of them showed strong nuclease activities (i.e., 2′,3′-cAMP and 2′,3′-cGMP synthetases) as reported[40] (Supplementary Fig. 16d), which hampered our assays to examine the deNAMing activity with in vitro transcribed NAD-RNAs. Although more investigation is definitely required, our results showed that the deNAMing activity of TIR domain-containing proteins might be evolutionarily conserved, at least in bacteria and archaea.

## Discussion

The existence of NAD-RNAs has been well-established in various organisms[5–16], uncovering a potential connection between NAD$^+$ metabolism and gene expression. Identification of the enzymatic machinery for adding or removing the NAD cap will help understand the biological functions of NAD-RNAs. Increasing studies over the past years have defined TIR domain-containing proteins as NADase enzymes that hydrolyze free NAD$^+$ in both prokaryotic and eukaryotic models. Here, our initial screening of putative NAD cap decapping enzymes led to the identification of TIR domain-containing proteins as NAD-RNA deNAMing enzymes, furthering our understanding of TIR domain's functionality. We demonstrated that AbTir, one of the bacterial TIR domain-containing proteins, can cleave in vitro synthesized and native NAD-RNAs using a combination of methods (Figs. 1 and 2), and v-cADPR-RNA was identified as the potential RNA product after cleavage (Fig. 3). However, whether v-cADPR-RNA exists in vivo, and if so, whether it is functional, is so far unknown. Our current study showed that NAD-RNAs can be first converted into cyclic ADPR-RNA by TIR domain-containing proteins, which can be further converted into p-RNA by Nudix or DXO/Rai1 decapping enzymes (Fig. 4c). This suggests the potential existence of a two-step NAD-RNA decapping strategy in vivo, which would be specific to NAD-RNAs due to the specificity of the deNAMing enzymes towards NAD-RNAs.

With our current results, only some of the bacterial and archaeal TIR domain-containing proteins showed deNAMing activity on NAD-RNA. It is unknown what features allow these TIR domain-containing proteins to possess such an activity. TIR domain-containing proteins are composed of versatile modules with distinct domain architectures across plants, animals, and prokaryotes[60]. However, we did not find any common architecture apart from the TIR domain among AbTir, BtpA, PdTir, and TcpA. We attempted to examine structural models of the investigated TIR domain proteins from both bacteria and Archaea. Interestingly, we found that all TIR domain-containing proteins with decapping activity (AbTir, BtpA, PdTir, and TcpA) tend to possess positive charges on the protein surface near the catalytic site, whereas

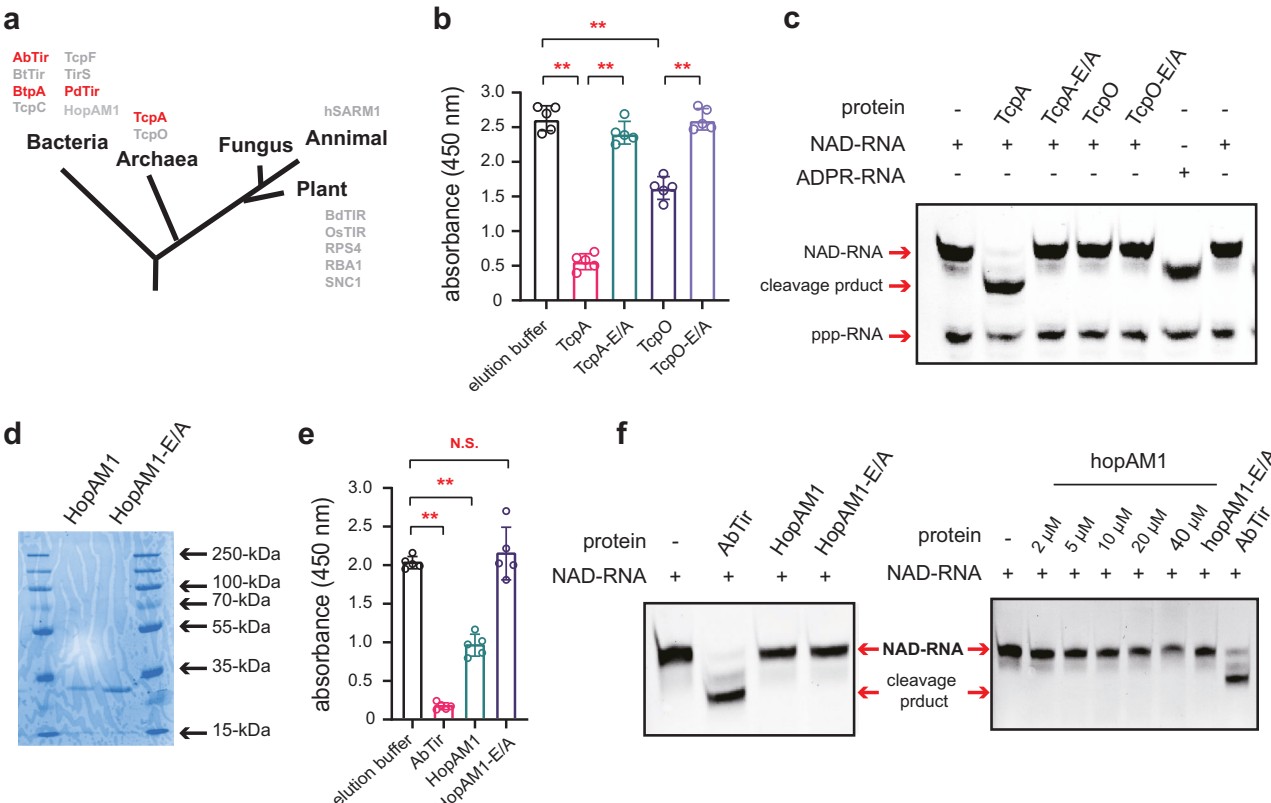

**Fig. 7 | Evaluation of the NADase and deNAMing acitivties of TIR domain-containing proteins in other organisms. a** The presence of typical TIR domain-containing proteins in all three domains of life. Red font labeled TIR domain-containing proteins have been confirmed to exhibit deNAMing activities in our study. **b** NADase activities of TcpA and TcpO from archaea. **c** An APB gel showing the RNA species after an in vitro transcribed NAD-RNA was incubated with TcpA or TcpO. In vitro transcribed ADPR-RNA and NAD-RNA were included as makers. **d** An SDS-PAGE gel showing the recombinant HopAM1 and HopAM1-E/A proteins purified from BL21 *E. coli* cells. The proteins were resolved in a 15% SDS-PAGE gel and visualized by Coomassie Brilliant Blue staining. **e** NADase activities of HopAM1. The

catalytic mutant HopAM1-E/A was included as a negative control. AbTir was included as a positive control. **f** APB gels showing the RNA species after reactions in which different concentrations of HopAM1 were incubated with an in vitro transcribed NAD-RNA. The reaction with AbTir was included as a positive control. The y-axes in (**b**, **e**) represent relative NAD+ levels, which were measured with the NAD/NADH Quantitation Kit by monitoring the absorption values at 450 nm. Error bars represent mean ± SD, which was calculated from five independent experiments ($n = 5$); [**] $P \leq 0.01$; N.S., not significant (calculated by the non-parametric Mann-Whitney U-test, two-sided). Exact $P$-values and source data are provided as a Source Data file.

TcpO -- the archaeal TIR domain protein without deNAMing activity -- exhibits a large block of negative charges (Supplementary Fig. 17). Considering the negative charges carried by RNA molecules, it is plausible that the TIR domain-containing proteins with deNAMing activity might have an affinity for RNA. However, the robust negative charges present on the TcpO surface may act to repel RNA molecules. It should be noted that biochemical experiments with purified proteins cannot recapitulate the true biologically active conformation of some TIR domain-containing proteins, especially for those from plants that have weak NADase activity and usually require an oligomerized complex to function. Thus, additional studies will be necessary to address whether TIR domain proteins in plants and animals have NAD-RNA deNAMing activity.

A plethora of recent studies underscored the importance of the NADase activity of TIR domain-containing proteins in immunity and proposed that NAD+ depletion or the generation of cleavage products might be a signaling event triggering downstream immune responses that lead to cell death[38,40,41,47,56]. Interestingly, nuclear localization of some plant TIR domain-containing proteins is required for disease resistance[62], which implies that TIR domain proteins might mediate gene expression regulation in the nucleus. Our study uncovered a novel enzymatic TIR domain function that can potentially reprogram the NAD-RNA transcriptome and raised the possibility that some TIR domain-containing proteins could regulate gene expression through NAD-RNA deNAMing.

## Methods

### Bacterial strains and culture media
*E. coli* DH5α and BL21 (DE3) strains used for plasmid construction and protein expression, respectively, were grown in Luria-Bertani (LB) medium amended with selective antibiotics at 37 °C. Ampicillin was included at a concentration of 50 μg/mL.

### Plasmids construction and PCR-mediated site-directed mutagenesis
Double-tagged (N-terminal MBP and C-terminal 6×His tags) TIR domain-containing proteins (full length or TIR domain-only) from bacteria, archaea, and plants were cloned into a modified pMAL-C2X empty backbone to prepare prokaryotic expression plasmids. Recombinant plasmids include MBP-AbTir-6×His (1-268aa), MBP-BtTir-6×His (1-287aa), MBP-TcpC^TIR−6×His (173-307aa), MBP-BtpA-6×His (1-275aa), MBP-TcpF-6×His (1-274aa), MBP-PdTir-6×His (1-299aa) and MBP-TirS-6×His (1-280aa) from Bacteria, MBP-TcpA-6×His (1-327aa) and MBP-TcpO-6×His (1-341aa) from archaea, RPS4^TIR+80aa (1-280aa) and RBA1 (1-362aa) from Arabidopsis, and BdTir (1-224aa) and OsTIR (1-196aa) from *Brachypodium distachyon* and *Oryza sativa*, respectively. DNA sequences optimized for prokaryotic protein expression were synthesized at Twist Bioscience and used as DNA templates for PCR amplification. The PCR products were gel-purified and confirmed by Sanger sequencing, and then ligated into the modified pMAL-C2X vector by Gibson Assembly cloning. PCR-mediated site-directed

mutagenesis was performed to produce the catalytic mutants, including MBP-AbTir-E/A-6×His, MBP-BtpA-E/A-6×His, MBP-PdTir-E/A-6×His, MBP-TcpA-E/A-6×His, and MBP-TcpO-E/A-6×His. All primers involved in constructing the above plasmids and generating the point mutations are listed in Supplementary Table 1.

## Recombinant protein expression and purification

The plasmids containing the TIR domains with the N-terminal MBP tag and the C-terminal 6 × His tag were transformed into competent *E. coli* BL21 cells. Single colonies were cultured in 1 mL LB containing 50 μg/mL ampicillin and grown overnight at 37 °C. These cultures were used to inoculate 400 mL of the same medium at 1:100 ratio. The large cultures were grown at 37 °C for 3-4 h until an OD value at 600 nm (OD600) of 0.4−0.6. Then, 0.2 mM isopropyl $\beta$-D-1-thiogalactopyranoside (IPTG; Thermo Fisher Scientific, 15529019) was added to induce protein expression, and the cells were grown at 16 °C for ~16 h. The bacterial cells were collected by centrifugation at 5000 g for 15 min at 4 °C. For protein purification, cell pellets were first resuspended in 20 mL of lysis buffer (100 mM Tris-HCl, pH 7.5; 500 mM NaCl; 5 mM 2-mercaptoethanol; 5% [v/v] glycerol; 20 mM imidazole; 0.1% [v/v] CA630) supplied with 1 mM protease inhibitor PMSF and 1× EDTA-free protease inhibitor cocktail. The cells were lysed by sonication, and the insoluble cell debris was removed by centrifugation at 10,000 g for 40 min. The recombinant proteins were purified with Profinity™ IMAC Resin, Ni-charged (Bio-Rad, 1560133). The proteins were eluted with the elution buffer (50 mM Tris-HCl, pH 7.5; 500 mM NaCl; 5 mM 2-mercaptoethanol; 5% [v/v] glycerol; 400 mM imidazole; 0.1% [v/v] CA630) and the concentrations were determined using the Bradford assay (Bio-Rad, Quick Start™ Bradford 1×Dye Reagent #5000205).

## Measurements of NAD⁺ levels in *E. coli* cells

The recombinant constructs of TIR domain proteins, as well as their corresponding catalytic E/A mutants, were transformed into competent *E. coli* BL21 cells. After confirmation by Sanger sequencing, single colonies were then cultured at 37 °C until they reached an OD600 of approximately 0.4−0.8, when IPTG was added to induce protein expression. The cultures at 2 h after IPTG induction were adjusted to OD600 = 0.5 ± 0.05, and the pellets from 500 μL of culture suspension were lysed with 200 μL 0.5 M perchloric acid (HClO₄). Samples were immediately placed on ice for 10 min and centrifuged at 12000 g for 10 min. 180 μL supernatant from each sample was transferred to a new tube, and 67 μL of 3 M K₂CO₃ was added. The reactions were placed on ice for another 10 min and centrifuged at 12000 g for 10 min. After centrifugation, 50 μL supernatant containing the extracted metabolites from each sample was mixed well with the Master Reaction Mix, which is a mixture of 2 μL NAD Cycling Enzyme Mix and 98 μL NAD Cycling Buffer from the NAD/NADH Quantitation Kit (Sigma-Aldrich, MAK037). The mixture was incubated for 5 min under dark condition at room temperature, allowing the reduction of NAD⁺ to NADH. Finally, 10 μL NADH developer from the NAD/NADH Quantitation Kit was added into each sample to react with NADH and a product that absorbs light at a wavelength of 450 nm was generated. The samples were transferred into a 96-well plate and measured for absorbance at 450 nm by spectrophotometry.

## RNA synthesis by in vitro transcription

A single-stranded DNA sequence T7φ2.5-A-32 (5′-CAGTAATACGACTCACTATTAGGCCTCTCGCTCTGCTGGGTGTGCGCTTGC-3′), which contains a T7 φ2.5 promoter at the 5′-end and a unique adenosine at the transcription start site, and its reverse complementary sequence (5′-GCAAGCGCACACCCAGCAGAGCGAGAGGCCTAATAGTGAGTCGTATTACTG-3′) were synthesized and annealed with each other to get the double-stranded DNA (dsDNA) template. In vitro transcription was carried out at 37 °C overnight with the reaction (100 μL) containing 2 μg of the dsDNA template, 1 × T7 polymerase buffer (New England

Biolabs), 1 mM CTP, 1 mM GTP, 1 mM UTP, and 1 mM ATP (for ppp-RNA) or NAD⁺ (for NAD-capped RNA) or FAD (for FAD-capped RNA) or dpCoA (for dpCoA-capped RNA) or ADPR (for ADPR-capped RNA) or Ap₄A (for Ap₄A-capped RNA), and 1 U/μL T7 RNA polymerase (New England Biolabs; M0251S). The RNA products were treated with DNase I (Roche) at 0.2 U/μL at 37 °C for 30 min, extracted by phenol/chloroform (5:1, pH 4.5), and then precipitated with ethanol. Unincorporated nucleotides were removed using Micro Bio-Spin P-30 Gel Columns (Bio-Rad) or by performing gel recovery after acryloylaminophenyl boronic acid affinity gel electrophoresis.

## Acryloylaminophenyl boronic acid (APB) affinity gel electrophoresis

APB affinity electrophoresis was employed for the purification and analysis of 5′-cap-modified RNAs[42]. To prepare the APB affinity gel, 80 mg 3-acrylamidophenylboronic acid (Boron Molecular, CAS: 99349-68-5) was dissolved into a mixture of the polyacrylamide solution (30 mL; including 1.5 mL 5 × TBE, 6.3 g urea, 5.76 mL 40% Acrylamide/Bis Solution) by rotating for 10 min. The solution was polymerized with APS (0.1% final concentration) and TEMED (0.1% final concentration). 0.5 × TBE was used as the gel running buffer for the APB gel. For the preparation of pure 5′-cap-modified RNAs, the in vitro transcribed RNA product was added to an equal volume of 2 × RNA loading dye (80% Formamide, 0.1% Xylene FF, 0.1% Brophenol Blue) and denatured at 65 °C for 5 min. After denaturing, the sample was immediately cooled down on ice for gel loading. The modified RNAs were collected by excising the gel and then eluted from the gel slice. For the analysis of the RNA products of various decapping reactions, the reactions were stopped by adding an equal volume of 2 × RNA loading dye and RNAs were denatured at 65 °C for 5 min. After APB gel electrophoresis, the gel was stained with ethidium bromide for 2 min on a shaker and visualized in the Molecular Imager® Gel Doc™ XR System.

## NADase and deNAMing/decapping assays

For NADase assays, the 20 μL reaction contained 20 μM NAD⁺, 10 μM purified proteins in NADase reaction buffer (50 mM Tris-HCl, pH 8.0; 100 mM NaCl; 20 mM MgCl₂), and 20% PEG6000. After incubation at 25 °C for various durations as specified in Results, 30 μL NAD/NADH extraction buffer from the NAD/NADH Quantitation Kit (Sigma-Aldrich, MAK037) was added to the reaction for measuring the NAD⁺ content. For RNA decapping assays, 500 ng in vitro transcribed RNAs (32-nt) capped with NAD, FAD, dpCoA, ADPR, or Ap₄A were incubated with the purified proteins (10 μM) in decapping buffer (50 mM Tris-HCl, pH 8.0; 100 mM NaCl; 20 mM MgCl₂; 20% PEG6000) at 25 °C for various durations as specified in Results. The reactions were mixed with an equal volume of 2 × RNA loading dye and RNAs were denatured at 65 °C for 5 min. The RNAs were resolved by running a denaturing 15% (w/v) APB gel with 0.5 × TBE as the running buffer and visualized by ethidium bromide staining.

## NAD cap detection and measurement with the NAD-capQ method

To measure the amount of the NAD cap before and after AbTir treatment, the NAD-capQ method was performed as previously described[43]. Briefly, in vitro transcribed NAD-RNA (1 μg) or total RNA (500 μg) isolated from *E. coli* (stationary phase) or Arabidopsis (12-day-old seedlings) were subjected to AbTir or AbTir-E/A treatment at 25 °C for 16 h. Then, the RNA products were extracted with phenol/chloroform (pH 4.5), precipitated, and subjected to nuclease P1 digestion in a 20 μL reaction with 1 × nuclease P1 Reaction Buffer and 10 U/μL nuclease P1 (New England Biolabs; M0660S) at 37 °C for 2 h. Before the AbTir cleavage reaction, the isolated total RNA was treated with NAP-10 columns to get rid of any residual free NAD⁺. Following digestion with nuclease P1, 30 μL of NAD/NADH Extraction Buffer from the NAD/

NADH Quantification Kit (Sigma-Aldrich, MAK037) was added to each sample. The 50 µL reactions were then used for the subsequent enzymatic cycling reaction and the colorimetric assay by following the manufacturer's protocol from the NAD/NADH Quantitation Kit. The same RNA samples treated with the AbTir-E/A catalytic mutant served as the negative controls.

### HPLC-MS analysis of various metabolites

Liquid chromatography was performed on a Nexera XR 40 series HPLC (Shimadzu) using a Synergi 4 µM Fusion-RP 80 Å 150 × 2 mm column (Phenomenex). The column temperature was kept at 40 °C, and the sample tray was maintained at 4 °C. 10 µL samples were injected at a flow rate of 0.2 ml/min with 10 mM ammonium formate (pH 4.2) and methanol as mobile phases A and B, respectively. Metabolites were eluted using the following gradient: 0–8 min, 8–90% B; 8–10 min, 90% B; 10–10.1 min, 90–8% B; 10.1–20 min, 8% B. The LCMS-8060 triple quadrupole mass spectrometer with electro spray ionization (Shimadzu) was operated in the positive mode. Scheduled multiple reaction monitoring (MRM) was employed to monitor analyte parent ions to product ion formation. MRM conditions were optimized using authentic standard chemicals including NAM ($[M + H]^+$ 123.00 > 80.00, 123.00 > 78.00, 123.00 > 53.00), $NAD^+$ ($[M + H]^+$ 664.00 > 136.00, 664.00 > 427.90, 664.00 > 523.95), cADPR ($[M + H]^+$ 541.80 > 136.15, 541.80 > 427.90, 541.80 > 347.90), and ADPR ($[M + H]^+$ 559.80 > 136.05, 559.80 > 427.95, 559.80 > 347.95). Data acquisition was performed using the LabSolutions LCMS v5.97 software, and data processing was carried out using LabSolutions Postrun (both Shimadzu). Metabolite products were quantified by scheduled MRM peak integration using calibration curves of standard chemicals.

### Cellular RNA isolation from *E. coli* BL21 strains and ribosomal RNA (rRNA) depletion

The culture of *E. coli* BL21 after overnight growth was diluted into fresh medium at a 1:100 ratio and the new culture was grown until an OD600 of 3.0 (the stationary phase). Isolation of total RNA was performed according to a published method[63]. In brief, cell pellets were collected by centrifugation at 6000 g for 10 min at 4 °C, and immediately put on ice for re-suspension by adding 20 mL TES buffer (10 mM Tris-HCl, pH7.5; 10 mM EDTA; 0.5% (w/v) SDS). The re-suspended cells were mixed with an equal volume of phenol/chloroform (5:1, pH 4.5) and incubated for 5–10 min at 65 °C with vigorous shaking. The cells were vortexed for 5 min at room temperature and centrifuged for 10 min at 12,000 g at 4 °C. The upper phase was collected and mixed with 20 mL of phenol/chloroform (5:1, pH 4.5) again. After another round of centrifugation (12,000 g, 4 °C), the upper phase was mixed with 10 ml chloroform. Then, the upper phase was gently collected after 10 min of centrifugation (12,000 g at 4 °C) and mixed with isopropanol at a 1:1 ratio. The mixture was incubated for 15 min at room temperature for RNA precipitation. Total RNA was pelleted by centrifugation (12,000 g at 4 °C) for 30 min, washed twice with 75% ethanol, and dissolved in DEPC $H_2O$.

Depletion of the 16S and 23S rRNAs was performed using the MICROBExpress™ Bacterial mRNA Enrichment Kit (Invitrogen, AM1905) according to manufacturer's instructions. The resulting rRNA-depleted RNA contains mRNA, tRNA, 5S rRNA, and other small RNAs and was used for subsequent experiments.

### Identification of NAD-capped RNAs by SPAAC-NAD-Seq

The rRNA-depleted RNA from 10 µg *E. coli* total RNA was used for preparing the SPAAC-NAD-Seq libraries according to ref. 15. At the same time, 100 ng of rRNA-depleted RNAs were used for regular RNA-Seq library construction as a control for NAD-RNA identification. NEBNext® Ultra™ II RNA Library Prep Kit for Illumina® (New England Biolabs, E7770) was employed to generate libraries for regular RNA-Seq and SPAAC-NAD-Seq with three biological replicates. After

sequencing, the clean data were first examined for the library quality with the FastQC program (https://github.com/s-andrews/FastQC). The data after quality check were subjected to a series of analyses, including low-quality nucleotide trimming, reads alignment, fragments counting, and differential gene expression analysis. In brief, the 3'-end low-quality nucleotides were trimmed with trim_galore (https://www.bioinformatics.babraham.ac.uk/). The remaining reads longer than 100-bp were mapped to the *E. coli* BL21 genome (https://www.genome.jp/kegg-bin/show_organism?org=ebl) using bowtie2 with default parameters. Next, HTSeq-count[64] was used to count reads that mapped to the genic regions. NAD-RNA-producing genes were identified with ratio of counts per million(CPM) between NAD-RNA-Seq and regular RNA-Seq ≥ 2 and FDR ≤ 0.05 using edgeR[65].

### Structural modeling

AlphaFold2[66] was used to predict the TIR domain structures of TcpA, BtpA, and TcpO. The predicted monomers of TcpA, BtpA, and TcpO were individually aligned onto the TIR domains of AbTir NADase asymmetric dimmer (7UXU). The PdTIR (3H16) protomer was also aligned onto the TIR domain of the AbTir NADase asymmetric dimmer. Since both the PdTIR structure and the predicted TIR structures of TcpA, BtpA and TcpO are in their inactive states, the BB-loops in these proteins present a closed conformation (Supplementary Fig. 17). Electrostatic potentials on the surface of proteins were visualized by PyMOL (v2.5) and color-coded, with red indicating negative charges, blue denoting positive charges, and white representing neutral charges.

### Statistics and reproducibility

Statistical analyses to examine significant differences and result visualization were performed with the GraphPad Prism 9 software. Significance was determined by $p < 0.05$ [*] or $p < 0.01$ [**]. All gel blots/plots were performed at least two times except for those in Fig. 5b.

### Reporting summary

Further information on research design is available in the Nature Portfolio Reporting Summary linked to this article.

## Data availability

The clean data of regular RNA-Seq and NAD-RNA-Seq in this study have been deposited in National Center for Biotechnology Information (NCBI) (BioProject database, accession code PRJNA1061094). All plasmids and strains supporting the finding of this study are available from the corresponding author upon request. Source data are provided with this paper.

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

## Acknowledgements

We would like to thank Dr. Liang Tong from Columbia University for his helpful discussions and suggestions. We also thank Dr. Ming Guo from University of Nebraska-Lincoln for generously sharing the expression plasmids of hopAM1 and hopAM1-E/A. This work was supported by grants from National Institutes of Health (NIH, GM061146) and Ministry of Science and Technology of the People's Republic of China (Chinese Ministry of Science and Technology, 2023YFC3402200) to X.C.

## Author contributions

X.W. and X.C. designed the project; X.W., D.Y., J.Y., H.H., R.H., Z. A., Q.C. and J.C. conducted laboratory experiments; X.W. performed the RNA-Seq data analysis and interpretation; X.C. supervised the project; X.W. and X.C. wrote the manuscript. All authors reviewed and revised the manuscript.

## Competing interests

The authors declare no competing interests.
