## [Peer Review File · Nature Communications]

Toll/interleukin-1 receptor (TIR) domain-containing proteins have NAD-RNA decapping activityREVIEWER COMMENTS

Reviewer #2 (Remarks to the Author):

The work mainly showed that TIR domain-containing proteins possess deNAMing activity towards NAD-capped RNAs and AbTir specifically produces a cyclic ADPR-RNA. The activities can be detected in vitro and when expressed in E.coli. The data support the claim. However, the deNAMing and cyclic ADPR synthetic activities of TIRs with NAD⁺ as substrate have been well characterized, especially for AbTir. So the paper basically just retested all the reported activities of TIRs on a NAD⁺ derivative (NAD-capped RNAs) in vitro and in E.coli without providing any biological data showing that if and how cyclic ADPR-RNA is functionally relevant. Hence the novelty and significance of this paper are not sufficient to be considered for publication in NC, and it is more suitable for a biochemical journal.

Reviewer #3 (Remarks to the Author):

The manuscript by Wang et al. presents a groundbreaking study on the discovery of a novel decapping mechanism involving NAD-RNAs mediated by TIR domain-containing proteins in bacterial and archaeal species. The authors demonstrate the removal of the NAM moiety from NAD-RNAs, leading to the production of cyclic ADPR-RNA. The study highlights the biological significance of this process in pathogen-host interactions. The findings significantly contribute to the understanding of noncanonical RNA capping and TIR-domain based sensing, shedding light on a previously unexplored area of research.

This is an exceptionally interesting and innovative work, bridging the gap between noncanonical RNA capping, sensing mechanisms, and pathogen-host interactions. The identification and characterization of a novel decapping enzyme, AbTir, add a valuable dimension to the existing knowledge in the field. The validation of the decapping mechanism in vivo provides a robust foundation for the study's conclusions. The experiments are meticulously conducted and described in great detail.

While I found the manuscript to be exceptionally well-prepared and a pleasure to read, there are a few minor remarks and suggestions that should be addressed before publication:

1. The authors are advised to correct the error in Line 43: "Streptomyces venezuelae (S. aureus)."
2. Figure Presentation: To enhance clarity, it is suggested to rearrange Figure 5 and Supplementary Fig. 8. Enlarging structural formulae could improve the visual representation.
3. The incomplete references (e.g., 20, 21, 41) need to be addressed.

Reviewer #4 (Remarks to the Author):

Wang et al uncover that TIR domain-containing proteins of three bacterial and one archaeal species can act on NAD-RNAs and propose a decapping role of these enzymes. TIR proteins are known to cleave free NAD into NAM and ADPR and earlier research identified a catalytic glutamate residue. Here, the authors demonstrate that some of these proteins can also cope with NAD-RNA substrates in vitro and that their heterologous production results in reduced NAD-RNA levels in Escherichia coli.

In general, most of the manuscript is easy to follow, images are clear and the text flows well. The results of two entire sections "Oligomerization enhances the deNAMing activity of AbTir" and "TcpA, a TIR domain-containing protein from Archaea, also exhibits deNAMing activity" are only represented by supplementary figures (Fig. S5 and FigS16, respectively) which is a bit surprising as the latter protein is also indicated in the abstract.

Major comments:

1) My main question after reading the manuscript is summarized in Fig. 4C by a red arrow indicating "function?". As all results were obtained in vitro or in a heterologous host, the biological significance of the observed activity cannot be evaluated and it is difficult to differentiate between (i) TIR domain-containing proteins that happen to be able to tolerate the attachment of an RNA moiety to their regular substrate and (ii) proteins that evolved to regulate NAD-RNA decapping. The authors conclude the abstract with a statement suggesting that the decapping activity of TIR proteins plays a role in pathogen-host interactions, but I do not see any support for this hypothesis in the text. A model (Fig. 4c) indicates that TIR proteins are effectively creating intermediates in an RNA decay pathway. The parallel existence of NAD-RNA decapping pathways without these intermediates poses the question of their importance. Is this pathway effective enough if in vitro conversion of NAD-RNA takes 2 to 16 hours (line 167)?

2) All investigated prokaryotic TIR proteins can act on NAD, but only some of them utilize NAD-RNAs. Is there a rational explanation for these observations? One could look at structure models of the investigated proteins to see if surface charge near the catalytic site plays a role. Additionally, investigation of the co-occurrence of the different NAD-RNA decapping pathways could be used to argue against (or for) functional redundancy.

3) The SPAAC-NAD-Seq results raise some questions. This protocol compares RNA-seq results that were treated to convert NAD-RNAs to biotin-RNAs with untreated RNAs and one should be careful about bias introduction. In vitro data suggests that almost all NAD-RNAs are targets of the AbTIR protein and only 12 NAD-RNAs remain intact. In vivo, however, only a "a small subset of genes involved in "molecule transport process" and "oxidoreductase activity" are targeted by AbTir". (...The authors should clarify that the mRNAs are targeted and not the genes). How do the authors explain these different observations and is there anything special about the few RNAs that are not decapped in vitro and the few RNA that are degraded in the heterologous host?

Minor comments:

1) Generation of in vitro substrates: It is not clear to me why cADPR-RNA was not obtainable by in vitro transcription (...or why this was indicated in the text (line 200)). In Fig. 4A, only a single substrate band should be visible as there is only one spot for non-canonical cap insertion. Are these caps not homogenous T7RNAP substrates?

2) Line 208: "Both ADPRC and CD38 are able to remove the NAM moiety from in vitro transcribed NAD-RNA, however, the identities of the RNA products have not been reported yet." - LC-MS analysis of CD38 cleavage products was performed (see. Fig. S12 in reference 24)

3) Fig. 3f: One peak is identified as "v-cADPR?" but in the text the authors state this (line 226) without the question mark. How was the identity of the peak verified?

4) Line 255: Recent identification of ADPR-RNA in eukaryotes and archaea should be considered.

5) References: Two references are from BioRxiv (ref 16, 53). Are there any updates that should be considered? Reference 21 does not have a journal title.

6) Fig. 6A "1. Collect cells..."

7) Fig. S7: This looks like raw gels of source data that can be cleaned up (i.e. place text outside of gels)

Dear Editors and Reviewers,

We sincerely appreciate your time and effort dedicated to reviewing our manuscript. It is great to learn that our work has been deemed potentially acceptable for publication in Nature Communications, pending adequate revisions. Your comments and suggestions have been insightful, and we have given each of them thorough consideration.

We are pleased to inform you that we have revised the manuscript and highlighted the changes in red for your convenience. Please find the updated version in the resubmitted files, along with our itemized responses provided below.

Thank you once again for your constructive feedback, and we look forward to any further guidance you may offer.

Best regards,

----- point-by-point responses -----

REVIEWER COMMENTS

Reviewer #2 (Remarks to the Author):

The work mainly showed that TIR domain-containing proteins possess deNAMing activity towards NAD-capped RNAs and AbTir specifically produces a cyclic ADPR-RNA. The activities can be detected in vitro and when expressed in E.coli. The data support the claim. However, the deNAMing and cyclic ADPR synthetic activities of TIRs with NAD⁺ as substrate have been well characterized, especially for AbTir. So the paper basically just retested all the reported activities of TIRs on a NAD⁺ derivative (NAD-capped RNAs) in vitro and in E.coli without providing any biological data showing that if and how cyclic ADPR-RNA is functionally relevant. Hence the novelty and significance of this paper are not sufficient to be considered for publication in NC, and it is more suitable for a biochemical journal.

Response: We thank the reviewer for finding that our experimental data support the conclusion regarding the deNAMing activity of TIR domain-containing proteins towards NAD-capped RNAs (NAD-RNAs). This conclusion is the primary message of our study. We respectfully disagree with the reviewer's assertion that our manuscript merely retested the reported activities of TIR domains on a NAD⁺ derivative. First, we contend that NAD-RNA is merely a NAD⁺ derivative. The attachment of the long RNA to NAD⁺ may influence the binding and cleavage capabilities of NADase proteins. Previous research has demonstrated that not all NADase proteins can perform NAD-cap cleavage. For instance, NudC, the initial NAD-RNA decapping enzyme discovered, exhibits robust NADase activity by hydrolyzing the pyrophosphate bond within NAD⁺ (Cahova et al., 2015, Nature). Conversely, NudE, another Nudix family protein, hydrolyzes free NAD⁺ but not NAD-RNAs as substrates (Abele et al., 2020, Biomolecules). Similarly, our findings indicate that not all TIR domain-containing proteins with NADase activity possess deNAMing capability on NAD-RNAs. Second, our research underscores distinctions between the cleavage products of ADPRC, CD38, and AbTir on free NAD⁺ and the deNAMing products on NAD-RNAs (Fig. 3 and Supplemental Figs.S5). This suggests that the deNAMing activity on NAD-RNA cannot be equated with the NADase activity on free NAD⁺, emphasizing potential differences in hydrolysis

mechanisms between free NAD⁺ and the NAD-cap.

Admittedly, providing biological data demonstrating the functional relevance of cyclic ADPR-RNA poses a challenge. However, this falls outside the scope of our current manuscript, which aims to report a novel decapping enzyme/pathway in prokaryotes and potentially other organisms. While we did not provide evidence regarding the existence and biological function of cleavage intermediates (ADPR-RNA, cADPR-RNA, or v-cADPR-RNA) *in vivo*, a recent Nature Communications publication reported the existence of ADPR-RNA in the archaeon *Sulfolobus acidocaldarius* and a specific ADPR-RNA decapping enzyme (*i.e.*, Saci_Nudt5), suggesting a potential role of ADPR-RNA in Archaea (Gomes-Filho et al., 2023, Nature Communications). However, the specific functions of NAD-RNAs and ADPR-RNAs remain largely unexplored, in part because research on these RNAs is still at an early stage.

We would like to reiterate the key findings of our manuscript and hope that the reviewer would appreciate the novelty and significance of this study. First, we observed that some TIR domain-containing proteins exhibit the ability to cleave the RNA NAD cap, releasing the NAM moiety (*i.e.*, the deNAMing activity named in our manuscript). Next, we identified a variant form of cyclic ADPR-RNA (v-cADPR-RNA) as the product resulting from treating *in vitro* transcribed NAD-RNA with AbTir. Moreover, we showed that TIR domain-containing proteins display decapping activity specific to NAD-RNAs, distinguishing them from other known decapping enzymes. Finally, the functional role of AbTir in bacterial cells was demonstrated by its inducible expression in *E. coli*, resulting in growth suppression and a significant decrease in NAD-RNA levels prior to effects on free NAD⁺. This confirms the *in vivo* deNAMing activity of TIR domain-containing proteins. Furthermore, we extend the deNAMing activity of TIR domains to Archaea, discovering that an archaeal TIR domain-containing protein, TcpA, also exhibits deNAMing activity on NAD-RNAs. Collectively, we believe our study holds importance and novelty, contributing substantially to the current knowledge of the NAD-RNA field. Given the widespread presence of TIR domain proteins in life, the implications of our research extend beyond the organisms studied, suggesting that NAD-RNA deNAMing may represent a conserved mechanism across different species.

Reviewer #3 (Remarks to the Author):

The manuscript by Wang et al. presents a groundbreaking study on the discovery of a novel decapping mechanism involving NAD-RNAs mediated by TIR domain-containing proteins in bacterial and archaeal species. The authors demonstrate the removal of the NAM moiety from NAD-RNAs, leading to the production of cyclic ADPR-RNA. The study highlights the biological significance of this process in pathogen-host interactions. The findings significantly contribute to the understanding of noncanonical RNA capping and TIR-domain based sensing, shedding light on a previously unexplored area of research.

This is an exceptionally interesting and innovative work, bridging the gap between noncanonical RNA capping, sensing mechanisms, and pathogen-host interactions. The identification and characterization of a novel decapping enzyme, AbTir, add a valuable dimension to the existing knowledge in the field. The validation of the decapping mechanism *in vivo* provides a robust foundation for the study's conclusions. The experiments are meticulously conducted and

described in great detail.

Response: We thank the reviewer for finding the study interesting and innovative. In response to the following questions raised by the reviewer, we have provided a point-by-point response below.

While I found the manuscript to be exceptionally well-prepared and a pleasure to read, there are a few minor remarks and suggestions that should be addressed before publication:

1. The authors are advised to correct the error in Line 43: “*Streptomyces venezuelae* (*S. aureus*).”

Response: Thank you for pointing out our mistake. We have rectified the mistake by changing “*Streptomyces venezuelae* (*S. aureus*)” to “*Streptomyces venezuelae* (*S. venezuelae*)” and have highlighted it in the revised manuscript (Line 44).

2. Figure Presentation: To enhance clarity, it is suggested to rearrange Figure 5 and Supplementary Fig. 8. Enlarging structural formulae could improve the visual representation.

Response: Thanks for the suggestions. Fig. 5 and Supplementary Fig. 8 have been updated as suggested by enhancing the clarity of the chemical structural formulae.

3. The incomplete references (e.g., 20, 21, 41) need to be addressed.

Response: We have updated the references accordingly as highlighted in the revised manuscript.

Reviewer #4 (Remarks to the Author):

Wang et al uncover that TIR domain-containing proteins of three bacterial and one archaeal species can act on NAD-RNAs and propose a decapping role of these enzymes. TIR proteins are known to cleave free NAD into NAM and ADPR and earlier research identified a catalytic glutamate residue. Here, the authors demonstrate that some of these proteins can also cope with NAD-RNA substrates in vitro and that their heterologous production results in reduced NAD-RNA levels in *Escherichia coli*.

In general, most of the manuscript is easy to follow, images are clear and the text flows well. The results of two entire sections “Oligomerization enhances the deNAMing activity of AbTir” and “TcpA, a TIR domain-containing protein from Archaea, also exhibits deNAMing activity” are only represented by supplementary figures (Fig. S5 and FigS16, respectively) which is a bit surprising as the latter protein is also indicated in the abstract.

Response: We appreciate your thoughtful review of our manuscript and the constructive suggestions. We agree with the valuable input and have incorporated essential components of Supplementary Fig. S16 into the main text, presenting them in a newly added Fig. 7. Considering the overall structure of the manuscript, we have opted to retain the results of the “Oligomerization enhances the deNAMing activity of AbTir” section in the supplemental file. This section serves to illustrate the optimization of AbTir's deNAMing activity. Thank you for your understanding and guidance in enhancing the clarity of our presentation.

Major comments:

1) My main question after reading the manuscript is summarized in Fig. 4C by a red arrow indicating “function?”. As all results were obtained in vitro or in a heterologous host, the

biological significance of the observed activity cannot be evaluated and it is difficult to differentiate between (i) TIR domain-containing proteins that happen to be able to tolerate the attachment of an RNA moiety to their regular substrate and (ii) proteins that evolved to regulate NAD-RNA decapping. The authors conclude the abstract with a statement suggesting that the decapping activity of TIR proteins plays a role in pathogen-host interactions, but I do not see any support for this hypothesis in the text. A model (Fig. 4c) indicates that TIR proteins are effectively creating intermediates in an RNA decay pathway. The parallel existence of NAD-RNA decapping pathways without these intermediates poses the question of their importance. Is this pathway effective enough if *in vitro* conversion of NAD-RNA takes 2 to 16 hours (line 167)?

Response: Thanks a lot for the reviewer's questions/comments. We agree that the biological significance of this decapping activity of AbTir is hard to be evaluated based on our current observations. Although we observed a decrease in cell proliferation in *E. coli* cells expressing AbTir compared to those expressing AbTir-E/A (Fig. 6b; Supplementary Fig. 12, a-c), it is challenging to definitively attribute this decrease to the consumption of free NAD⁺ or alterations in the NAD-RNA transcriptome. Identifying a specific amino acid(s) to differentiate the NADase activity from the decapping activity is crucial but has yet to be done. Nevertheless, our observation that the change in NAD-RNAs precedes that in free NAD⁺ (Fig. 6) is noteworthy. Given that not all TIR domain-containing proteins exhibit decapping ability on NAD-RNAs, we tend to think that these TIR proteins possess NAD-RNA decapping activity as the result of their sequence or structural properties (see response to point 2 below).

For the statement in the abstract, we realized that it was overstated and have modified the description in the abstract and introduction sections (Line 33 and Line 107). Recently, TIR domain-containing proteins were frequently reported to be involved in bacterial, animal, and plant immunity, which prompted us to raise the implication that TIR domain-mediated NAD-RNA decapping might be a potential mechanism in pathogen-host interactions. We view this as an exciting direction for future exploration. Given the lack of data in this respect, we removed such statements in Abstract and Introduction.

For the proposed model in Fig. 4c, we thank the reviewer for raising an intriguing question. Currently, our evidence limits us from making definitive statements about the significance of this decapping pathway. Hence, we put a question mark there in the figure. In a previous study published in *Nature Chemical Biology*, Grudzien-Nogalska et al. (2019) reported the co-existence of the DXO deNADding enzyme and the Nudix protein Nudt12 decapping enzyme in mammalian cells. Their study revealed functional roles for these enzymes on distinct NAD-RNA targets. Furthermore, they observed selective modulation of NAD-RNA levels by either Nudt12 or DXO when cells were exposed to nutritional or environmental stress conditions (Grudzien-Nogalska et al. 2019, *Nature Chemical Biology*). These findings propose that the coexistence of parallel pathways engaged in NAD-RNA decapping may confer a cellular advantage by enabling specific regulation of diverse NAD-RNA targets in response to varying environmental conditions.

As of now, we do not have information on the existence and biological function of cleavage intermediates (ADPR-RNA, cADPR-RNA, or v-cADPR-RNA) *in vivo*. A recent publication in *Nature Communications* reported the existence of ADPR-RNA in the thermophilic model archaeon

Sulfolobus acidocaldarius, with a specific ADPR-RNA decapping enzyme (Saci_Nudt5) identified (Gomes-Filho et al., 2023, Nature Communications). None of the recombinant Nudix proteins from *S. acidocaldarius* exhibited NAD-RNA decapping ability, suggesting potential utilization of different pathways to process NAD-RNAs in this organism. In our current study, the discovery that TcpA, an archaeal TIR domain-containing protein, exhibits deNAMing activity on NAD-RNAs supports the possibility of a TIR-catalyzed NAD-RNA decapping model *in vivo*. However, the specific roles of NAD-RNAs and ADPR-RNAs remain undiscovered, and further investigations are required to determine the presence of ADPR-RNAs in other organisms.

2) All investigated prokaryotic TIR proteins can act on NAD, but only some of them utilize NAD-RNAs. Is there a rational explanation for these observations? One could look at structure models of the investigated proteins to see if surface charge near the catalytic site plays a role. Additionally, investigation of the co-occurrence of the different NAD-RNA decapping pathways could be used to argue against (or for) functional redundancy.

Response: Thank you for your insightful question and suggestions. Indeed, with our current findings, only certain bacterial and archaeal TIR domain-containing proteins have deNAMing activity on NAD-RNAs. The features that enable these particular TIR domain-containing proteins to possess such an activity remain unknown (as discussed in Discussion section of our previous manuscript). Prompted by this reviewer's comments, we examined structural models of the investigated TIR domain proteins. Interestingly, we found that all TIR proteins with decapping activity (AbTir, BtpA, PdTir, and TcpA) tend to possess positive charges on the surface near the catalytic site, whereas TcpO -- the archaeal TIR domain protein without deNAMing activity -- exhibits a large block of negative charges (see the figure below). Considering the negative charge carried by RNA molecules, it is plausible that the TIR domain-containing proteins with deNAMing activity might have an affinity for RNA. However, the robust negative charges present on the TcpO surface may act to repel RNA molecules. We have added this result as Supplementary Fig. 17 and the corresponding description in the Discussion section of the revised manuscript.

Figure Legend: The cryo-EM structure of AbTir asymmetric dimer (7UXU) is shown in surface representation, with electrostatic potential mapped to the surface. The 3' position of 8-amino-isoquinoline adenine dinucleotide (3AD) bound in the catalytic center is labeled. The TIR

domain structures of TcpA, BtpA and TcpO were predicted by AlphaFold2 (Jumper et al. 2021). TcpA, BtpA and TcpO and PdTIR (3H16) were superimposed onto the AbTir asymmetric dimer. Blue and red colors indicate the positive and negative charges mapped on the surface of proteins.

3) The SPAAC-NAD-Seq results raise some questions. This protocol compares RNA-seq results that were treated to convert NAD-RNAs to biotin-RNAs with untreated RNAs and one should be careful about bias introduction. In vitro data suggests that almost all NAD-RNAs are targets of the AbTIR protein and only 12 NAD-RNAs remain intact. In vivo, however, only a “a small subset of genes involved in “molecule transport process” and “oxidoreductase activity” are targeted by AbTir”. (…The authors should clarify that the mRNAs are targeted and not the genes). How do the authors explain these different observations and is there anything special about the few RNAs that are not decapped in vitro and the few RNA that are degraded in the heterologous host?

Response: Thanks for the reviewer’s comments and questions. SPAAC-NAD-Seq has been proven to be a sensitive and accurate method to identify NAD-capped RNAs, presenting several advantages over NAD captureSeq (Hu et al. 2021, PNAS). Hu et al. performed experiments to validate the authenticity of NAD-RNAs identified by SPAAC-NAD-Seq. Thus, we directly adopted SPAAC-NAD-Seq in our experiments. By comparing the identified NAD-RNAs in our study with the previous results from NAD tagseq II conducted in a different lab, most NAD-RNAs were repeatedly detected (see Supplementary Fig. 10a). This further underscores the accuracy and robustness of our SPAAC-NAD-Seq technology.

Regarding the differing observations of AbTir targets between *in vitro* and *in vivo* experiments, we think these are two distinct experiments that cannot be compared. In the *in vitro* experiment, rRNA-depleted *E. coli* RNA was pre-treated with wild-type AbTir or catalytically inactive AbTir-E/A under 25°C for 30 minutes. Under these conditions, only 12 NAD-RNAs were detected after AbTir treatment, using the criteria of "NAD-RNA-Seq/regular RNA-Seq ≥ 2 and FDR ≤ 0.05 ." It is conceivable that extending the reaction time or using a higher protein concentration might result in the decapping of all NAD-RNAs. In fact, all 12 remaining NAD-RNAs exhibited lower enrichment levels in the wild-type AbTir pre-treatment sample compared to the catalytically inactive AbTir-E/A pre-treatment sample. In the *in vivo* experiment, rRNA-depleted *E. coli* RNA was isolated from AbTir- or AbTir-E/A-expressing cells after 15 minutes of IPTG induction, during which AbTir or AbTir-E/A protein levels were likely quite low (the proteins can only be detected by western blotting, not by Coomassie staining; Figure 6e). The selection of this time point, when free NAD⁺ levels showed no significant change between AbTir- and AbTir-E/A-expressing cells, was based on the rationale that NAD⁺ can be cotranscriptionally incorporated into the RNA 5'-end as a NAD cap during transcription initiation, and maintaining constant NAD⁺ content would minimize differential NAD-RNA production. Thus, both *in vitro* and *in vivo* experiments provide supporting evidence for the decapping activity of AbTir. However, it is not appropriate to directly compare the deNAMing targets. Another difference between the *in vitro* and *in vivo* experiments (in addition to the amount of AbTir) is that the RNAs were denatured in the *in vitro* experiment, which might have helped to “expose” the NAD cap what may be otherwise inaccessible due to RNA structures *in vivo*.

To clarify that the mRNAs are targeted and not the genes, we have modified the description of

the “Introduction” section in the revised manuscript (Line 101).

Minor comments:

1) Generation of *in vitro* substrates: It is not clear to me why cADPR-RNA was not obtainable by *in vitro* transcription (•••or why this was indicated in the text (line 200)). In Fig. 4A, only a single substrate band should be visible as there is only one spot for non-canonical cap insertion. Are these caps not homogenous T7RNAP substrates?

Response: Thanks for the good questions. In our manuscript, we tried to obtain ADPR-RNA and cADPR-RNA by *in vitro* transcription with the same DNA template (see Materials and Methods section) using ADPR and cADPR (ADPR, Sigma-Aldrich A0752; cADPR, Sigma-Aldrich C7344) in place of ATP, respectively. However, despite the successful generation of ADPR-RNA, obtaining cADPR-RNA proved unsuccessful under the same conditions. We hypothesize that the cyclic linkage of the C:N bond might impact the efficiency of cADPR incorporation during transcription initiation.

Regarding the presence of two bands in the *in vitro* transcription products, it is correct that there is only a single spot in the DNA template that should direct the insertion of the non-canonical cap. Additionally, all cap substrates are claimed to have a purity of $\geq 90\%$ (HPLC). Therefore, one would anticipate observing only a single band corresponding to the non-canonically capped RNA. However, in our experiments, we consistently observed a second band during *in vitro* transcription. We suspect that the lower band might be 5'-ppp-RNA with the initiating A nucleotide skipped during transcription. This inference is supported by its resistance to direct digestion by "Terminator 5'-Phosphate-Dependent Exonuclease" but susceptibility to digestion after pre-treatment with "RNA 5' Polyphosphatase" that converts 5'-ppp-RNA to 5'-p-RNA (see the Figure below).

2) Line 208: “Both ADPRC and CD38 are able to remove the NAM moiety from *in vitro* transcribed NAD-RNA, however, the identities of the RNA products have not been reported yet.” – LC-MS analysis of CD38 cleavage products was performed (see. Fig. S12 in reference 24)

Response: We appreciate the reviewer for bringing up this point. The previous study asserted the verification of ADPR-RNA from the cleavage products of CD38 on *in vitro* transcribed NAD-RNA through MALDI-MS analysis (Abele et al., 2020, Biomolecules). However, it is important to note that the authors focused solely on confirming the existence of ADPR-RNA, without examining other potential RNA products such as cADPR-RNA and v-cADPR-RNA. There is a possibility that, in

addition to ADPR-RNA, other cleavage products may have been generated by CD38 on NAD-RNAs. In our study, we confirmed the generation of both ADPR-RNA and cADPR-RNA from the CD38 cleavage reaction on NAD-RNA. Even so, we have realized that our previous description was not accurate, and have accordingly revised the sentence in the revised manuscript (Line 210).

3) Fig. 3f: One peak is identified as “v-cADPR?” but in the text the authors state this (line 226) without the question mark. How was the identity of the peak verified?

Response: We apologize for any confusion in presenting Fig. 3f in the previous manuscript. Indeed, we should have removed the “?” in the figure. In a prior study, it was demonstrated that AbTir, functioning as a NADase enzyme, can cleave free NAD⁺ into NAM and a cADPR isomer known as v-cADPR or 2'-cADPR (Manik et al., 2022, Science). Since v-cADPR or 2'-cADPR is not commercially available, we opted to use the product of AbTir on free NAD⁺ as a standard to identify the deNAMing product of AbTir on NAD-RNAs. As anticipated, our analysis of the reaction product of AbTir on free NAD⁺ by HPLC-MS revealed a peak distinct from ADPR or cADPR and is most likely v-cADPR (given that the three product ions of cADPR can be detected simultaneously; Supplementary Fig. 6). Consequently, we believe that the detected product after the deNAMing reaction of AbTir on NAD-RNA is v-cADPR-RNA, as illustrated in Fig. 3f. To prevent any misunderstanding, we have omitted the “?” in the revised manuscript.

4) Line 255: Recent identification of ADPR-RNA in eukaryotes and archaea should be considered.

Response: Yes, we totally agree with the reviewer’s point that the recent identification of ADPR-RNA in Archaea should be considered (Gomes-Filho et al. 2023, Nature Communications). We have modified and added more description about this in the revised manuscript. To our knowledge, ADPR-RNA has not yet been identified in eukaryotes. Please see the detailed edits in Line 254-257.

5) References: Two references are from BioRxiv (ref 16, 53). Are there any updates that should be considered? Reference 21 does not have a journal title.

Response: Thanks for your reminder. Yes, both papers (ref 16 and 53) have been officially published online recently. We have duly updated them, along with reference 21, as highlighted in the revised manuscript.

6) Fig. 6A “1. Collect cells…”

Response: We are sorry for our carelessness and have corrected the typo in the revised Fig. 6.

7) Fig. S7: This looks like raw gels of source data that can be cleaned up (i.e. place text outside of gels)

Response: We have placed the annotated text outside the gels based on the reviewers’ suggestion.

References:

Abele, F. *et al.* A novel NAD-RNA Decapping pathway discovered by synthetic light-up NAD-RNAs. *Biomolecules* 10, 513 (2020).

Cahová, H. *et al.* NAD captureSeq indicates NAD as a bacterial cap for a subset of regulatory RNAs. *Nature* 519,

374-377 (2015).

Gomes-Filho, J. *et al.* Identification of NAD-RNA species and ADPR-RNA decapping in Archaea. *Nature Communications* 14, 7597 (2023).

Manik, M. *et al.* Cyclic ADP ribose isomers: Production, chemical structures, and immune signaling. *Science* 377, eadc8969 (2022).

Hu, H. *et al.* SPAAC-NAD-seq, a sensitive and accurate method to profile NAD⁺-capped transcripts. *Proceedings of the National Academy of Sciences* 118, e2025595118 (2021).

Jumper, J. *et al.* Highly accurate protein structure prediction with AlphaFold. *Nature* 596,583-589 (2021).

Grudzien-Nogalska, E. *et al.* Structural and mechanistic basis of mammalian Nudt12 RNA deNADding. *Nature Chemical Biology* 15, 575–582 (2019).

REVIEWERS' COMMENTS

Reviewer #4 (Remarks to the Author):

I thank the authors for their detailed response to my comments. The manuscript convincingly shows that some TIR proteins cannot only cleave free NAD into NAM and ADPR, but also utilize NAD-RNA molecules as substrates. The biological relevance of this observation remains unknown and it is not clear if cyclic ADPR-RNA are important functional molecules. As a consequence, the authors removed overstated claims that include possible roles of TIR proteins in pathogen-host interactions.

The authors have added a supplementary figure (Fig. S17) that provides a rational explanation for the presence of NAD-RNA discrimination. In future experiments, this interesting observation could be validated via mutational analysis of differentially charged regions.

The authors satisfactorily address most of my comments and provide also a plausible explanation for the presence of multiple bands during in vitro capping experiments. In one instance, they state that "To our knowledge, ADPR-RNA has not yet been identified in eukaryotes". Here, Weixler I et al Nucleic Acids Res. 50, 2022) have previously identified ADPR-RNAs in human cells.

Reviewer #4 (Remarks to the Author):

I thank the authors for their detailed response to my comments. The manuscript convincingly shows that some TIR proteins cannot only cleave free NAD into NAM and ADPR, but also utilize NAD-RNA molecules as substrates. The biological relevance of this observation remains unknown and it is not clear if cyclic ADPR-RNA are important functional molecules. As a consequence, the authors removed overstated claims that include possible roles of TIR proteins in pathogen-host interactions.

The authors have added a supplementary figure (Fig. S17) that provides a rational explanation for the presence of NAD-RNA discrimination. In future experiments, this interesting observation could be validated via mutational analysis of differentially charged regions.

The authors satisfactorily address most of my comments and provide also a plausible explanation for the presence of multiple bands during in vitro capping experiments. In one instance, they state that “To our knowledge, ADPR-RNA has not yet been identified in eukaryotes”. Here, Weixler et al (*Nucleic Acids Res.* 50, 2022) have previously identified ADPR-RNAs in human cells.

Response: We appreciate the reviewer's positive feedback. As noted by the reviewer, Weixler et al. (2022) identified ADPR-RNA (ADP-ribosylated RNA) in mammalian cells, introducing ADPr as a novel non-canonical RNA-cap (Weixler *et al.*, 2022). However, ADP-ribosylation is catalyzed by a family of proteins called PARPs and occurs at the 5'-terminal phosphate of RNA. The resulting ADP-ribosylated RNA (also referred to as ADPR-RNA, as depicted in Figure 1c) exhibits structural differences from the ADPR-RNA generated by RNA polymerase or by TIR domain cleavage of NAD-capped RNA (shown in Figure 1a, b). Please take note of the distinction in the linkage bond at the ADPR-cap (encircled in the figure). Thus, we did not cite this paper in our current manuscript.

*This figure (Figure 1) is modified from a review paper (Wiedermannova *et al.*, 2021).

References:

- Weixler, L., Feijs, K. L. H. and Zaja, R. (2022) ADP-ribosylation of RNA in mammalian cells is mediated by TRPT1 and multiple PARPs. *Nucleic Acids Res*, **50**, 9426-9441.
- Wiedermannova, J., Julius, C. and Yuzenkova, Y. (2021) The expanding field of non-canonical RNA capping: new enzymes and mechanisms. *R Soc Open Sci*, **8**, 201979.